# Embryo mechanics cartography: inference of 3D force atlases from fluorescence microscopy

Sacha Ichbiah[1], Fabrice Delbary[1], Alex McDougall[2], Rémi Dumollard ⬤[2] & Hervé Turlier ⬤[1]✉

Tissue morphogenesis results from a tight interplay between gene expression, biochemical signaling and mechanics. Although sequencing methods allow the generation of cell-resolved spatiotemporal maps of gene expression, creating similar maps of cell mechanics in three-dimensional (3D) developing tissues has remained a real challenge. Exploiting the foam-like arrangement of cells, we propose a robust end-to-end computational method called 'foambryo' to infer spatiotemporal atlases of cellular forces from fluorescence microscopy images of cell membranes. Our method generates precise 3D meshes of cells' geometry and successively predicts relative cell surface tensions and pressures. We validate it with 3D foam simulations, study its noise sensitivity and prove its biological relevance in mouse, ascidian and worm embryos. 3D force inference allows us to recover mechanical features identified previously, but also predicts new ones, unveiling potential new insights on the spatiotemporal regulation of cell mechanics in developing embryos. Our code is freely available and paves the way for unraveling the unknown mechanochemical feedbacks that control embryo and tissue morphogenesis.

Understanding the mechanical regulation of embryo and tissue shape emergence is a long-standing goal in developmental biology and biological physics. Although gene expression patterning in early embryos is increasingly documented thanks to recent single-cell sequencing methods[1], we still know very little about how cellular forces are spatiotemporally patterned within embryos and tissues. This is due to the lack of efficient methods for extracting cell- and time-resolved mechanics in a systematic, tissue-wide and noninvasive manner.

Most experimental methods to measure mechanics[2] are local and time-consuming, such as micropipette aspiration, atomic force microscopy measurement or embedded droplet deformation[3–12], making the generation of spatiotemporal maps of mechanics tedious; others are invasive, such as laser ablation, perturbing normal tissue development[13,14], or they probe mechanics only at the tissue level[15–17],

ignoring mechanical heterogeneities within the multicellular structure. All methods require live 3D imaging to follow the deformation of cells, tissues or embedded objects. Advances in fluorescence microscopy allow us to record the geometry of cells during the development of an embryo in toto from the zygote to a few hundreds of cells with a confocal microscope[18] and up to thousands of cells with a light sheet microscope[19]. Attractive new microscopy techniques have emerged to try to quantify cellular mechanics directly, such as Brillouin microscopy[20], or membrane tension probes[21,22], but such methods still lack cross-validations and remain difficult to link directly to mechanical models of tissues.

An alternative idea that emerged a decade ago is to infer the forces that dictate the shape of cells directly from their geometry by solving an inverse mechanical model[23]. These mechanical inference

[1]Center for Interdisciplinary Research in Biology, College of France, CNRS, INSERM, University of PSL, Paris, France. [2]Laboratory of Developmental Biology of the Villefranche-sur-Mer, Institute of Villefranche-sur-Mer, Sorbonne University, CNRS, Villefranche-sur-Mer, France. ✉e-mail: herve.turlier@college-de-france.fr

methods (also called force or stress inference) are based only on image analysis and do not require tissue perturbation: they have therefore a lower entry barrier than many other methods, as they do not require complex experimental setups. They have been shown to be efficient in inferring tensions (and pressure) in two-dimensional (2D) cell monolayers[24] and can be scaled to hundreds or thousands of cells. For tissues and embryos, most inference methods are based on the hypothesis that cells adopt shapes and arrangements similar to bubbles in a foam, as pointed out by D'Arcy Thompson more than a century ago[25]. This analogy implies that the mechanics of cells is dominated by tensile stresses on their surface, which are generated by actomyosin contractility[26]. Because actomyosin contractility may be regulated differentially in distinct cells or at different interfaces (such as the cell–medium and cell–cell interfaces[5]), embryos and tissues may be seen as heterogeneous foams, where each cellular interface may adopt a different tension. Actual inference methods also assume generally a quasistatic mechanical equilibrium, where the viscous relaxation of tensions (dozens of seconds) is much faster than typical developmental time scales (dozens of minutes to hours). This foam-like mechanical equilibrium underpins two force balances, the Young–Dupré and Young–Laplace equations ('Results'), relating surface tensions with contact angles and cell pressures with interface curvatures. In the following, we will therefore refer to tension and pressure inference.

First versions of tension inference methods[27,28] neglected Laplace's law by assuming straight cell interfaces, as in traditional vertex models[29]. In addition, they treated tensions and pressure as independent variables, which made the inverse problem generally underdetermined and relatively sensitive to noise. Alternatively, segmentation of cell membranes into 2D polygonal lines to explicitly measure their curvature[30,31] allows successive determinations of tension and pressure and makes the set of equations generally overdetermined. In the particular case where the whole tissue can be imaged with its boundaries, as this is generally the case for early embryos, the problem turns out to be systematically overdetermined. However, the generalization of this approach to three dimensions has not been convincing so far, since high-quality images and a robust segmentation pipeline are required[32,33]. To avoid such issue, an elegant variational 2D approach was recently proposed in which cell junctions are fitted by circular arcs to find tensions and pressure[34], taking advantage of a mapping between a heterogeneous 2D foam and the tiling of the space into 'circular arc polygons'. This tiling falls actually within the class of Möbius diagrams[35], whose mapping to 2D foams has already been pointed out mathematically[36]. In three dimensions, however, interfaces have mean constant curvatures but are generally not portions of a sphere and may adopt saddle-node shapes, as already noted for homogeneous foams[37]. Contrary to a recent assumption in ref. 38, the generalization to 3D of the variational scheme developed by ref. 34 with Möbius diagrams is mathematically not correct.

To fill the gap, we propose foambryo[39], a robust end-to-end computational method for performing tension and pressure inference in three dimensions, starting directly from 3D fluorescence microscopy of cell membranes. Our pipeline follows the 2D approach of ref. 30, where we decouple the inferences of tensions and pressures. It relies particularly on a new and efficient surface mesh reconstruction method to precisely quantify cell geometry. Our inversion algorithm exploits furthermore junction lengths and interface areas as weights to infer tensions and pressures more robustly. We perform a comprehensive benchmarking of our pipeline using 3D foam-like mechanical simulations and a systematic sensitivity analysis on various equivalent force balance formulas. Our inference pipeline yields convincing results on early embryos of mice, worms and ascidians by recovering known mechanical characteristics and predicting new ones. We provide an easy-to-install Python package and a comprehensive set of user-friendly 3D visualization tools.

## Results

### Delaunay-watershed algorithm for multimaterial mesh generation

An essential first step is to extract the precise geometry of cells from microscopy images. Voxel-based segmentation masks are heavy data structures that are not well adapted to measuring geometrical features such as contact angles or mean curvatures. Alternatively, triangle mesh representations of cell interfaces possess several advantages; they are sparse data structures that facilitate the retrieval of geometric quantities using a discrete differential formulas[40]. They are easy to render graphically and form basic elements for computational modeling, such as vertex models[41,42] or finite element methods[43]. The surface meshes of interest in our case are triangular, nonmanifold to account for tricellular junctions and multimaterial to keep track of the identity of each enclosed cell or region ('material'), in the spirit of ref. 44. Although triangle meshes can be generated by discretizing voxel-based segmentation masks directly, using marching cube algorithms[45] or more recent methods[46], we found that previous algorithms introduced large errors in angle measurements in general.

Therefore, we developed a new algorithm that robustly generates nonmanifold multimaterial surface meshes from cell segmentation masks (preexisting or obtained with cellpose[47]). The first step consists of computing from the cell segmentation masks a Euclidean distance transform map (EDT)[48], which represents a smooth topographic map of cell (and image) boundaries (Fig. 1a). This EDT map may also be predicted directly from raw fluorescent images by training a convolutional neural network[49]. From the distance map, we sample points at the extrema of the elevation value using a maximum-pooling operator, which serves as control points to generate a Delaunay tessellation of the space (triangulation in 2D or tetrahedralization in 3D). A dual Voronoi diagram is then generated from the Delaunay tessellation and is represented as an edge-weighted graph $\mathcal{G} = (\mathcal{N}, \mathcal{E}, \mathcal{W})$, where $\mathcal{N}$ is the set of nodes, representing tetrahedra in the dual space (triangles in 2D), $\mathcal{E}$ the set of edges between these nodes and $\mathcal{W}$ their associated weights. These weights are defined here according to the average value of the integrated distance map measured along the corresponding triangle (or edge in 2D) in the dual space (Extended Data Fig. 1a). Seeding each region using masks, we partition this graph using a watershed algorithm[50] that separates the nodes in the graph between the different cells and the external region. Other graph partitioning methods such as multicut[51], hierarchical agglomeration[52] or Mutex watershed[53] algorithms may also be envisioned, although we have not tried them here. Mapped back on the dual Delaunay space, this partition defines a unique surface (contour in 2D) mesh that accurately follows cell boundaries. Our Delaunay-watershed mesh generation algorithm works just as well in 2D as in 3D (Fig. 1a). Since the main purpose of this mesh generation algorithm is to extract precise geometrical features, we generated a set of 47 foam-like simulations of embryos with a number of cells varying from 2 to 11, which we translated into artificial confocal fluorescent images of a size of roughly ($250 \times 250 \times 250$ pixels) (ref. 54) to compare the error generated for different geometrical measures of interest (contact angles, mean curvature, junction length, area and volume) by our pipeline and state-of-the-art surface meshing techniques implemented in CGAL[46]. Our Delaunay-watershed algorithm[55] outperforms CGAL[46] for the retrieval of contact angles (Fig. 1b), and cell volumes or junction lengths (Extended Data Fig. 1b), while its precision is comparable for the retrieval of interface areas and mean curvatures (Extended Data Fig. 1b).

### Tension and pressure balance

Once the geometry of the cells can be calculated from the cell segmentation mesh (Extended Data Fig. 2 and Supplementary Note), we have to formulate the inverse mechanical problem to retrieve the relative force of the cells from their geometry. A quasistatic foam-like equilibrium

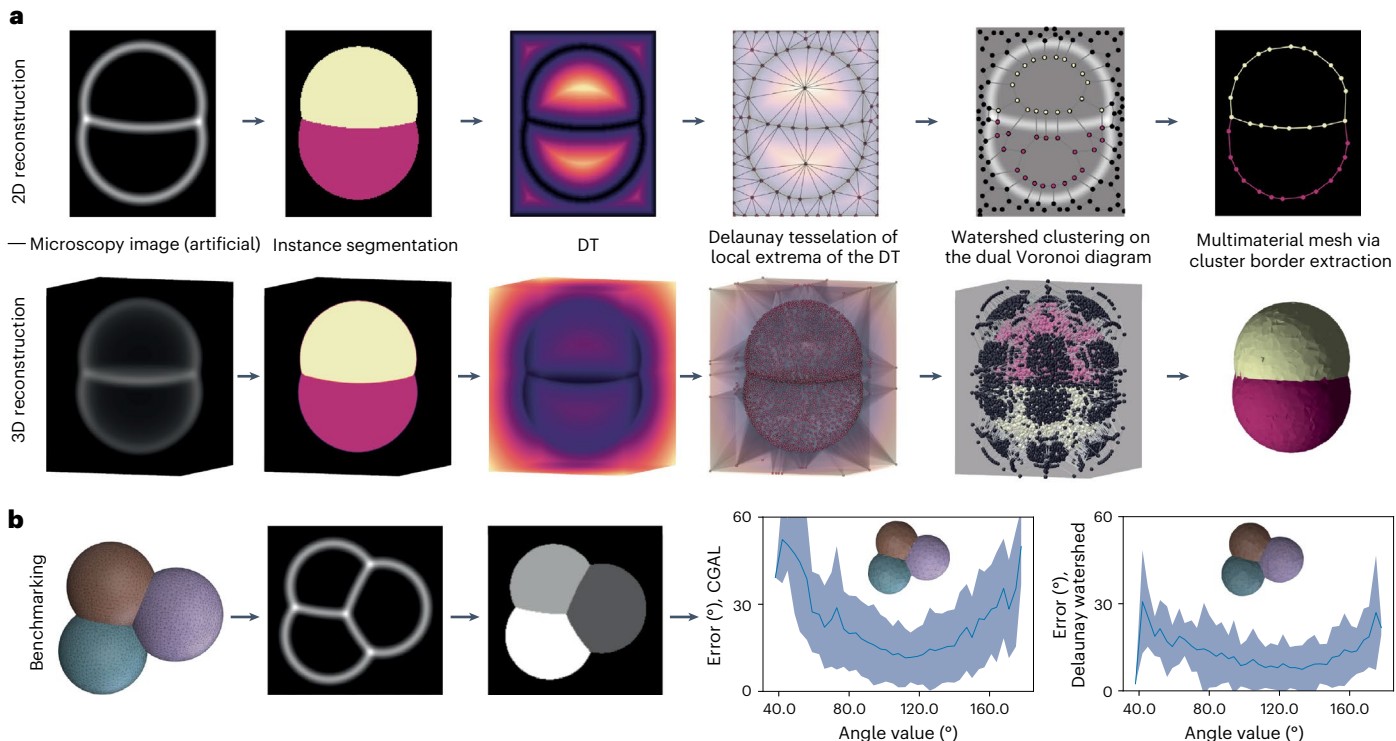

**Fig. 1 | Multimaterial mesh generation algorithm. a**, Description of the successive steps of the Delaunay-watershed algorithm in 2D and 3D. From left to right: microscopy image (artificial here), segmentation mask, EDT map, seeding from extrema of the EDT and Delaunay tessellation, dual Voronoi diagram after the graph was segmented with a watershed cut algorithm[50] and multimaterial nonmanifold mesh (polygonal in 2D or triangle mesh in 3D). DT, distance

transform. **b**, The geometric precision of our mesh generation algorithm is benchmarked on foam-like simulations ($n = 54$), which are transformed into artificial images to reconstruct surface meshes. Our pipeline reconstructs cell geometry with better precision than state-of-the-art mesh generation methods, such as CGAL[46], as shown by the comparison of the error (mean ± s.d.) in the reconstructed angles as a function of the original angle.

underpins two stress balance equations within the tissue (Fig. 2d). The Young–Laplace equation

$$p_i - p_j = \gamma_{ij} H_{ij} \qquad (1)$$

relates the hydrostatic pressure difference $p_i - p_j$ between cells of indices $i$ and $j$ with the interface tension $\gamma_{ij} = \left\| \boldsymbol{\gamma}_{ij} \right\|$ and the interface mean curvature $H_{ij}$, which is homogeneous along each interface (by convention the index 0 will refer to the external medium). The Young–Dupré force balance

$$\boldsymbol{\gamma}_{ij} + \boldsymbol{\gamma}_{jk} + \boldsymbol{\gamma}_{ki} = \mathbf{0} \qquad (2)$$

states that the sum of vectorial tensions should be zero at each tricellular junction line that joins the interfaces between cells $i$, $j$ and $k$. This vectorial sum is equivalent to saying that tensions are coplanar and form a triangle, which implies the triangle inequality $\gamma_{ij} < \gamma_{jk} + \gamma_{ki}$ and equivalent relations by permutation of the indices $i$, $j$ and $k$. Noncompliance with one of these inequalities indicates that tension balance breaks down and predicts generically a topological transition in the embryo or tissue. The Young–Dupré tension balance can generically be decomposed into a set of two independent scalar equations that combine the polar angles between the interfaces $\alpha_{ij}$, $\alpha_{jk}$ and $\alpha_{ki}$ (Fig. 2d). In the following, we use five different variants of tension balance that involve cosines and sines of polar angles only[23], which we named Young–Dupré, Young–Dupré projection, Lami, inverse Lami and Lami logarithm[56] (Methods and Supplementary Note).

　The balance of forces in a foam-like tissue of $n_c$ cells can also be derived from the minimization of surface energy under cell volume constraints. This formulation is particularly adapted to numerical

simulations on a discrete mesh[41,42] and is based on a Lagrangian function that includes pressures as Lagrange multipliers enforcing volume conservation. Based on derivatives of this Lagrangian function with respect to vertex position, we defined two additional expressions of tension and pressure balances, that we named variational Young–Dupré and variational Laplace (Supplementary Note).

**Tension and pressure inference**

Tensions depend only on contact angles at tricellular junctions and are independent of cell pressures, so here we decompose the inverse problem into two steps, in the same spirit as ref. 30. First, we solve the tensions and then determine the cell pressures using inferred tension values. The advantage of this two-step approach is that tensions can still be inferred in embryos or tissues under confinement or compression (such as *Caenorhabditis elegans*), where Laplace's force balance does not apply anymore, since the interfaces may adopt nonuniform mean curvatures. Tensions (and pressures) are known up to a multiplicative (respectively, an additive) factor. To remove this indeterminacy, we impose that the average tensions shall be equal to unity, which adds an equation to the system, and we arbitrarily fix the external pressure to zero. The tension inference problem can be generically cast into a linear system $A_\Gamma \times \Gamma = c_\Gamma$, where $\Gamma = (\gamma_1 \ldots , \gamma_m)^T$ (T denoting a transpose) collects the $n_m$ unknown tensions, $A_\Gamma$ is a matrix of size $(n_\Gamma + 1) \times n_m$ that collects $n_\Gamma + 1$ equations that relate the tensions and $c_\Gamma = (0, \ldots , 0, n_m)^T$ implements the constraint on the average tensions. This system is generally overdetermined and is solved in the sense of ordinary least-squares. Performing a systematic benchmark of our method, we found that better results are obtained when the $n_\Gamma$ tension equations are weighted by the length of the corresponding junction (Supplementary Note), which is the choice taken further.

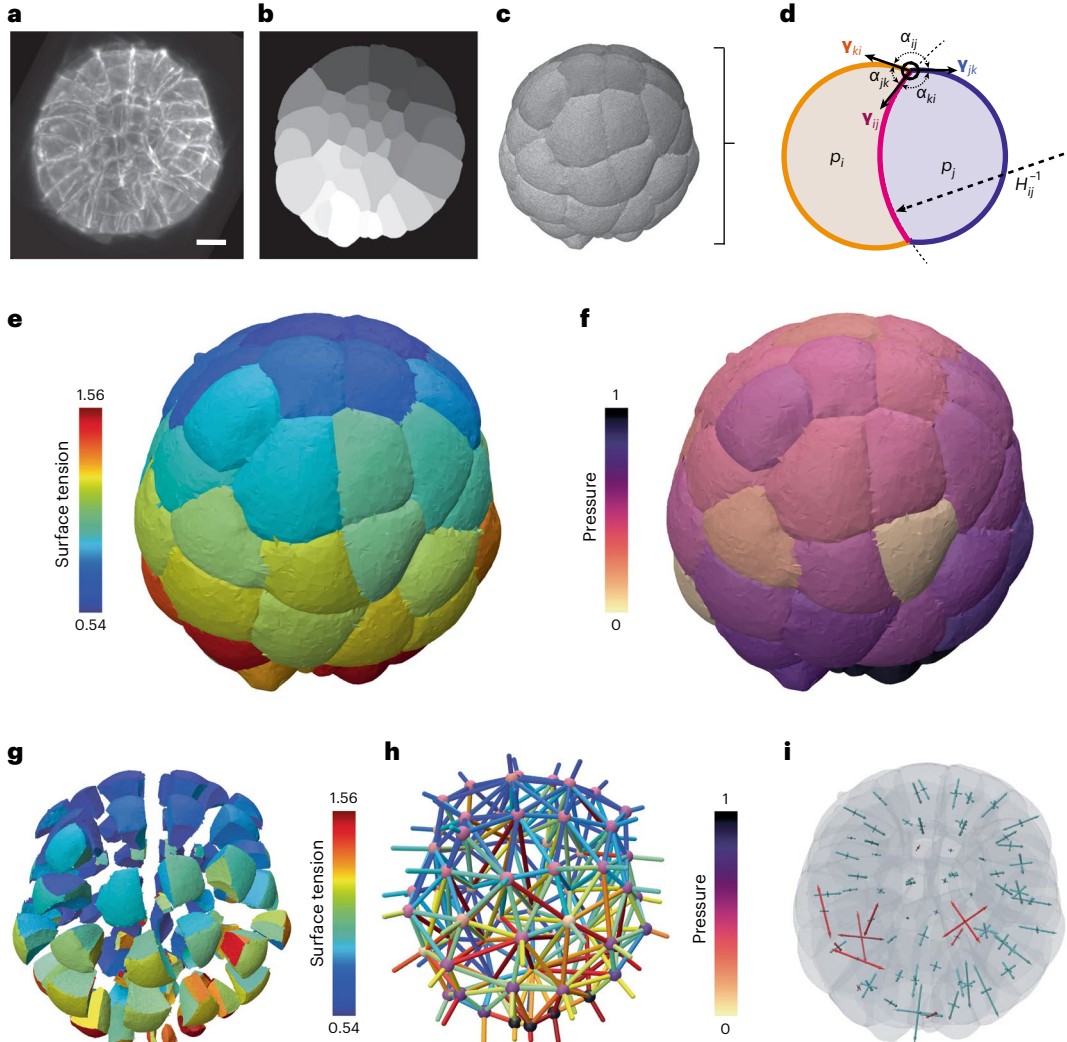

**Fig. 2 | 3D force-inference procedure and resulting mechanical atlas for a 64-cell ascidian embryo. a**, 3D fluorescence microscopy image (maximum projection) of a 64-cell *P. mammillata* embryo (from ref. 19). Scale bar, 20 µm. **b**, Cell segmentation mask in one focal plane of the 3D image. **c**, Multicellular surface mesh of cell interfaces. **d**, Schematic cell doublet illustrating the two force balances that need to be inverted: the Young–Dupré equation that relates surface tensions $\gamma_{ij}$, $\gamma_{ik}$ and $\gamma_{jk}$ with contact angles $\alpha_{ij}$, $\alpha_{ik}$ and $\alpha_{jk}$, and the Young–Laplace equation that relates cell pressure difference $p_j - p_i$ with tension $\gamma_{ij}$ and the radius of the interface curvature $H_{ij}^{-1}$. **e**, 3D map of relative surface tensions in the embryo, plotted with a color code from blue (lowest) to red (highest). **f**, Pressure map in the embryo, normalized from 0 to 1. **g**, Exploded view of the surface tension map that illustrates cell–cell contact tensions within the embryo. **h**, Force graph representation of the mechanical atlas, where each node represents a cell with its associated pressure and each edge corresponds to an interface colored by its tension value. **i**, 3D stress eigenvalue representation, corresponding to a stress tensor calculated per cell with the Batchelor formula[71]. Positive eigenvalues are plotted in blue (compressive stress) while negative are plotted in red (extensile stress).

In Fig. 3, we compare the sensitivity of our inference algorithm for the different variants of the Young–Dupré formula in equation (3), and the variational Young–Dupré equation. By perturbing vertex positions with random noise in mesh solutions of foam-like simulations (Fig. 3a), we calculate and plot the mean square error on the tensions inferred from this perturbed mesh (Fig. 3b). At low noise values, we find that the scalar Young–Dupré equation gives better results, but this error increases faster for larger noise. Variational Young–Dupré and the different Lami variants have an error that increases faster at low noise, but then reaches a lower relative plateau at higher noise.

For pressure inference, we follow the same approach, expressing the inverse problem as a linear system $A_P \times P = B_P$ where $P = (p_1, \dots, p_c)^T$ collects the unknown pressures, which we solve with the ordinary least-squares method. Here we compare the traditional Laplace equation (1) and our new variational Laplace formula (Supplementary Note). We find that our mesh-based variational

formula performs systematically better regardless of the level of noise (Fig. 3c).

Error in inference results may originate from deviations of cells shape from the solution of an heterogeneous foam or from an insufficient image resolution (Extended Data Fig. 3c), but they are also the result of an inevitable intrinsic noise generated by our pipeline that comes from the segmentation and meshing operations. To evaluate which formula may be most adapted given this minimal and ineluctable level of noise, we generate ideal artificial confocal microscopy images from mesh results of foam-like simulations (Supplementary Note). This dataset[54] is used to benchmark our method: the images are segmented using cellpose[47] and translated into multimaterial meshes with our Delaunay-watershed algorithm to ultimately infer tensions and pressures using the various formulas introduced earlier (Fig. 3d). In general, we find that the systematic error induced intrinsically by our pipeline remains low (below 10% on average), with the best inference

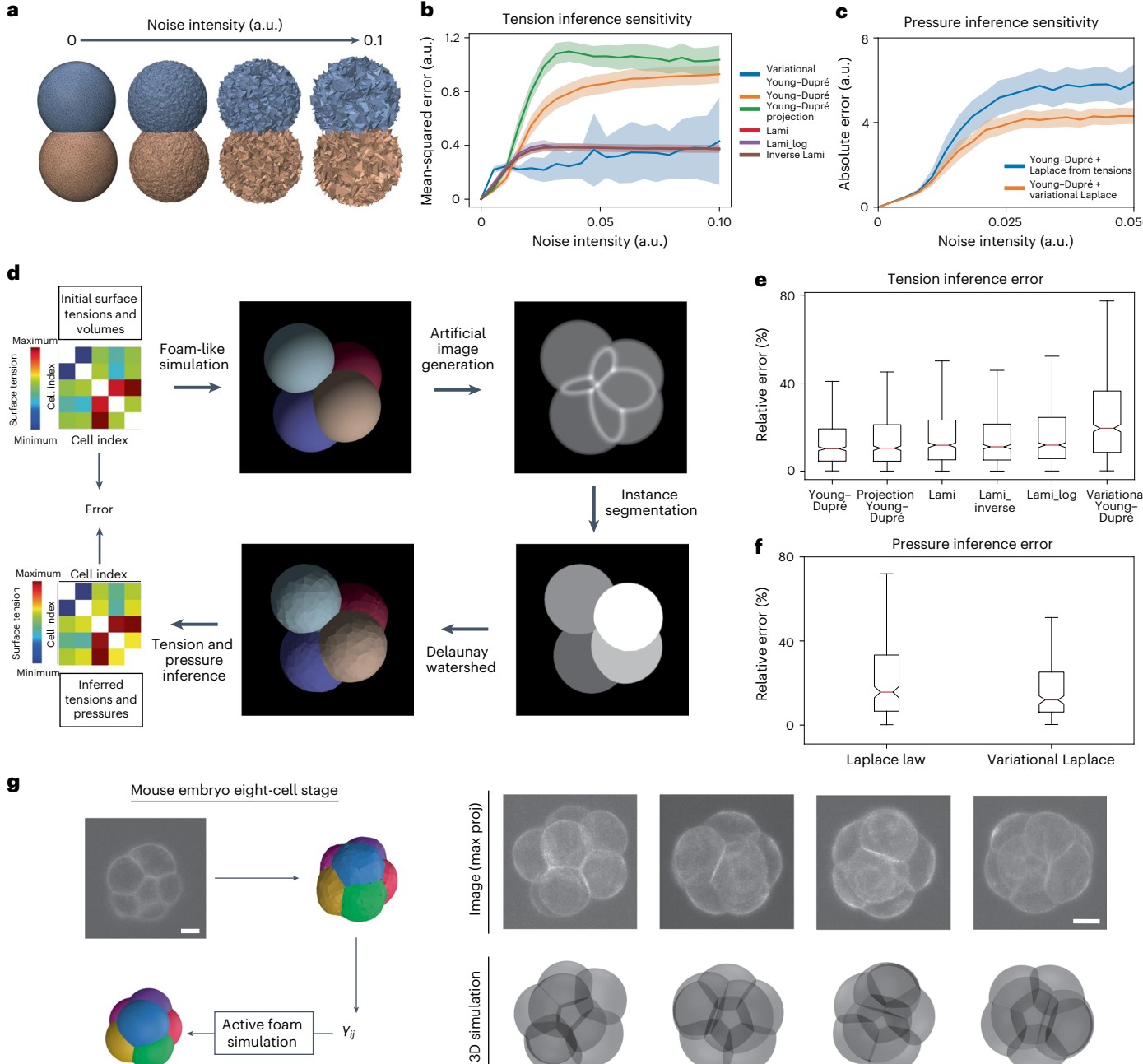

**Fig. 3 | In silico validation of the force-inference pipeline. a,** Sensitivity analysis of different formulas for tension and pressure inference. Foam-like simulation meshes are perturbed by randomly displacing each vertex position following a uniform law. **b,** Plot of the mean squared error on inferred tensions (mean ± s.d.) as a function of the intensity of the noise for the different tension formulas ($n$ = 54 simulated embryos with 2 to 11 cells). **c,** Plot of the absolute error (mean ± s.d.) on the inferred pressure as a function of the noise intensity for the Laplace and variational Laplace formulas ($n$ = 54 simulated embryos). **d,** Pipeline for benchmarking force inference: from random surface tensions (visualized here as a symmetrical $(n_c + 1) \times (n_c + 1)$ matrix) and cell volume values, foam-like embryo meshes are simulated, from which artificial microscopy images are generated; then, our end-to-end pipeline is applied to regenerate a mesh and infer tension and pressure values. Min, minimum and Max, maximum. **e,** Plot of the relative error in the inferred tensions for the different tension inference formulas applied to our simulated embryo dataset ($n$ = 54 simulated embryos; the red center line denotes the median value, while the box contains the 25th to 75th percentiles of the dataset; the whiskers mark the 5th and 95th percentiles). **f,** Relative errors on inferred pressures on our simulated embryo dataset with Laplace and variational Laplace formulas ($n$ = 54 simulated embryos; the red center line denotes the median value, while the box contains the 25th to 75th percentiles of the dataset; the whiskers mark the 5th and 95th percentiles). **g,** Self-consistent validation of the inference on the compaction of the eight-cell mouse embryo. Surface tensions are inferred with the pipeline and averaged between the cell–medium and cell–cell interfaces. Foam-like simulations are performed using these tensions and yield an in silico embryo morphology that is compared to the real embryo image. Scale bars, 20 µm.

results obtained with the scalar Young–Dupré equation (3) and the variational Laplace formula (Fig. 3e–f and Extended Data Fig. 3b), consistent with the histograms of eigenvalues of the corresponding pseudo-inverse matrices (Extended Data Fig. 3a). For all tension and pressure inference examples shown below, we therefore systematically use the scalar Young–Dupré and variational Laplace formula.

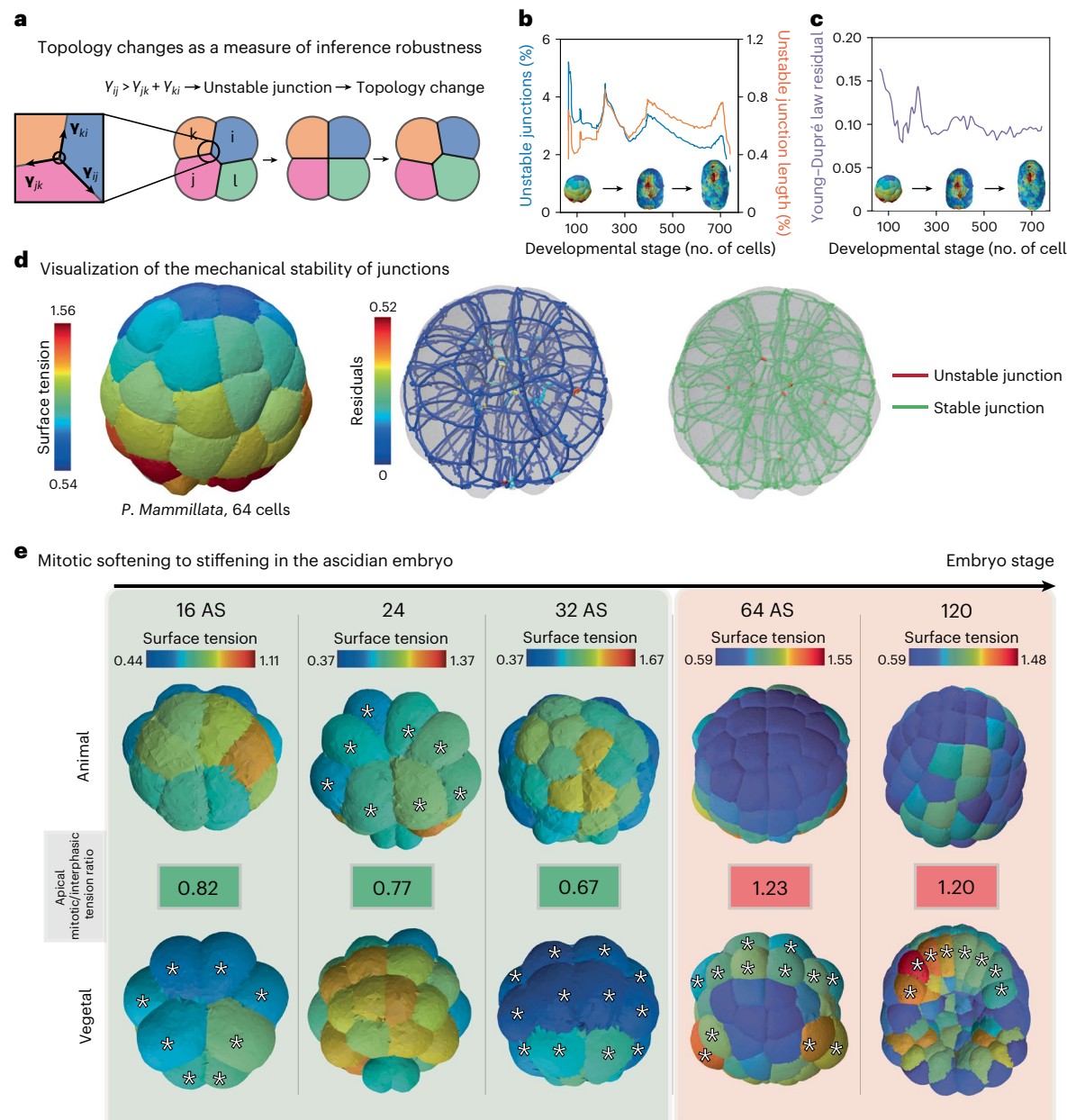

**Fig. 4 | In vivo validation of the 3D tension inference. a**, Illustration of the process of T1 topological transition when one tension at a junction becomes greater than the sum of the two others. **b**, Plot of the percentage of unstable junctions in the embryo (blue) and the ratio of unstable junction length to total junction length in the embryo as a function of its development stage, defined by its number of cells. **c**, Plot of the mean residual of Young–Dupré equations (purple) in the embryo as a function of its development stage, defined by its number of cells. **d**, The left shows the surface tension map of the 64-cell ascidian embryo. The middle shows a visualization of the residuals of Young–Dupré equations for each junction in the same embryo. The right shows the junctions for which inference predicts a T1 topological transition are mechanically unstable. Visualization of stable (green) and unstable (red) junctions in a 64-cell ascidian embryo (*P. mammillata*). **e**, Maps of apical tension at the animal and vegetal poles of the early ascidian embryo (*P. mammillata*) in the 16AS (AS for asynchronous), 24-, 32AS, 64- and 120-cell stages. The ratio of mitotic to interphase apical tension is colored green if it is less than one and red if it is greater than one. Mitotic cells are indicated with a white star.

## Force inference applied to early embryo development

To validate the biological relevance of our new force-inference pipeline, we inferred 3D mechanical atlases of mouse and ascidian embryos using fluorescent microscopy images of cell membranes. We first study the self-consistency of the heterogeneous foam model in compacting eight-cell mouse embryos. Compaction corresponds to the extension of internal cell contacts that round up the embryo and was shown by micropipette tension measurements[5] to be characterized by a decrease in the ratio $\alpha = \frac{\gamma_{cc}}{2\gamma_{cm}}$, called the compaction parameter, where $\gamma_{cm}$ is the tension at the cell–medium interface of cells and $\gamma_{cc}$ the tension at the cell–cell contacts. This single parameter is enough to characterize the embryo shape and is equal to the cosine of half the contact angle of the cell medium. Using confocal fluorescent images of eight-cell mouse embryos at successive levels of compaction, we segmented them into multimaterial meshes and inferred relative tensions. We then performed 3D foam-like simulations and compared them with the original microscopy images (Fig. 3g), and found a very good qualitative agreement. Our automatic inference methods yields systematically lower variability in inferred $\gamma_{cc}$ values than previously obtained by measuring contact angles manually[5], as illustrated on Extended Data Fig. 3d.

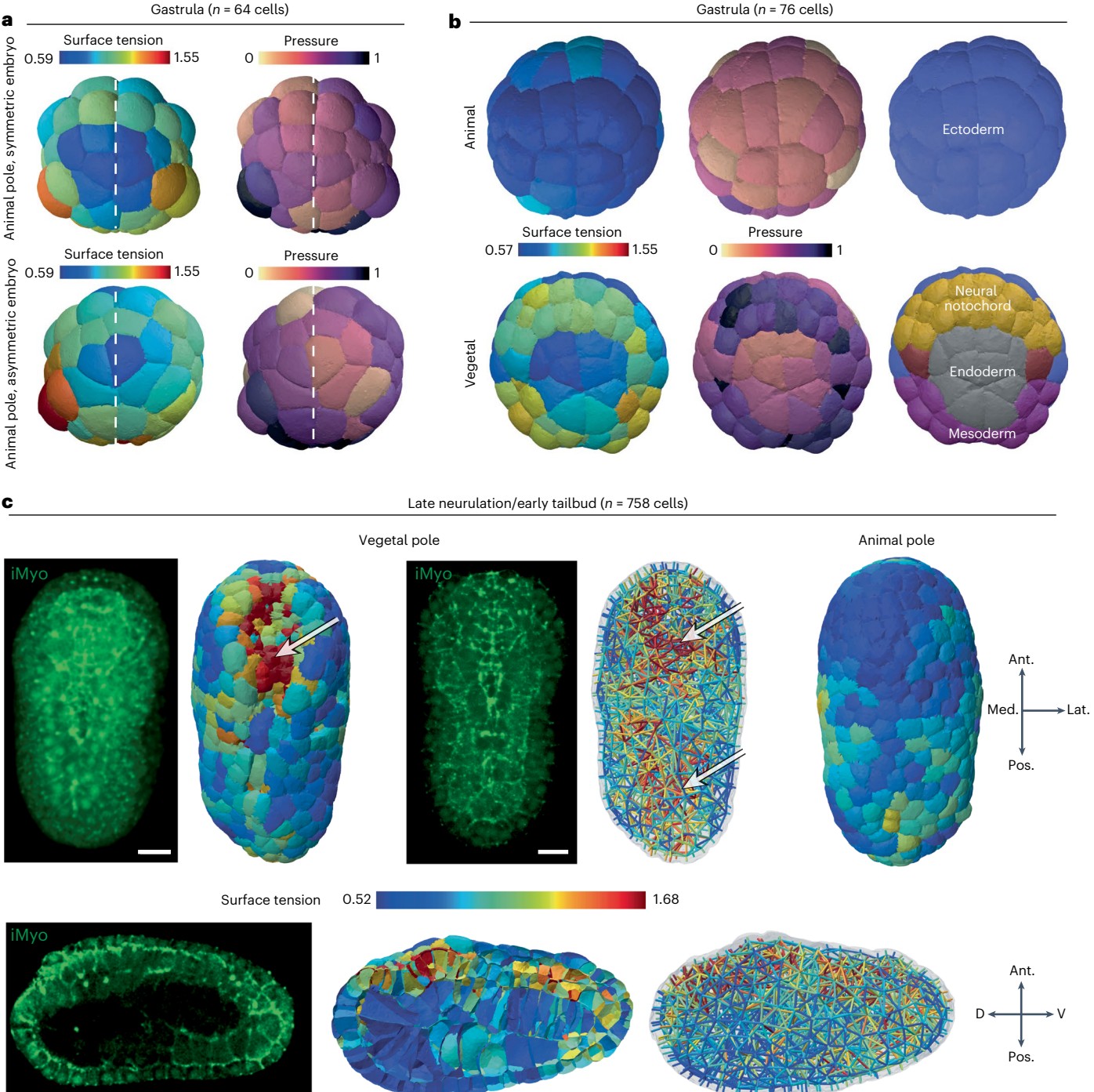

**Fig. 5 | Spatiotemporal patterning of mechanics in the ascidian embryo *P. mammillata*. a**, Tension and pressure maps of the animal pole of two 64-cell embryos. Imperfections in the geometric bilateral symmetry of the embryo are reflected by a corresponding asymmetry in the apical tension and pressure of the cell. **b**, Tension and pressure maps at the animal and vegetal poles of a 76-cell embryo and the corresponding pattern of cell fate in the germ layers. **c**, Tension maps in late neurula (758 cells) from vegetal, animal and sagittal views.

The white arrows indicate regions of higher tension (red) within the embryo. Representative fluorescent microscopy images of myosin II (iMyo) at the vegetal pole and in the sagittal view: 3D reconstruction (top left) or selective plane projection of ten confocal planes (top middle and bottom left). The orientation of the embryo is given by arrows. Ant., anterior; Pos., posterior; Med., medial; Lat., lateral; D, dorsal and V, ventral. Scale bars, 20 μm.

This confirms the relevance of a heterogeneous foam model hypothesis and exemplifies the capability of our inference pipeline.

To go beyond this example, where cell–medium and cell–cell tensions are uniform within the embryo, we inferred spatiotemporal mechanical atlases of the early ascidian embryo *Phallusia mammillata*. We used fluorescent images of cell membranes that were acquired with

a confocal microscope from the zygote to the 44 cell stage (Methods) or with a light sheet microscope from the 64-cell stage to the late neurula (less than or equal to 800 cells)[19]. We first focused on the shape of the embryo from 16 cells to the early gastrula, where divisions are reported to be asynchronous with cell divisions that alternate between the animal and vegetal hemispheres[57]. Recently, it was shown in *P. mammillata*

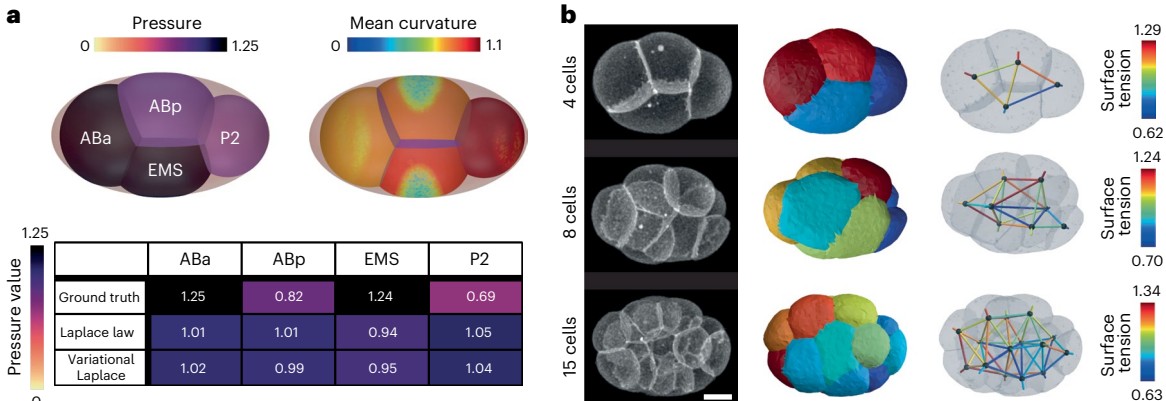

**Fig. 6 | Force inference in the *C. elegans* embryo. a**, The top shows the pressure inference on a simulated embryo confined in a rigid shell leading to heterogeneous mean curvatures patterns at cell–medium interfaces. The bottom shows a comparison of exact and inferred pressures (Laplace and variational Laplace formula). **b**, Inference of tension only on the *C. elegans* embryo at 4, 8 and 15 cell stages from microscopy images (source ref. [18]). Scale bar, 10 μm.

embryos at 16, 32 and 44 cell stages that cells at mitosis entry have lower apical tension than their interphase counterparts located in the opposite hemisphere[6]. This striking result, in notable contrast to mitotic cortical stiffening reported in most somatic cells[58], is accurately predicted by our force-inference method, which finds a ratio of apical tension between mitotic and interphase cells that is systematically lower than 1 in the 16 to 32 cell stages (Fig. 4e). This mitotic softening alternates between the animal and vegetal poles, as also illustrated in pressure maps (Extended Data Fig. 4a), further explains the overall 3D shape of the embryo that is flatter on the side of interphase cells (16 and 32 cells). As one would expect from Laplace's law, if applied globally to the embryo approximated to a droplet, a higher apical tension at one pole leads indeed to its flattening. Inference not only confirms previous results, but also predicts an unknown switch in the 64-cell embryo where mitotic blastomeres have higher apical tension than their interphase neighbors (Fig. 4e and Extended Data Fig. 4a) suggesting that, from this stage on, cells undergo mitotic stiffening. This mitotic stiffening persists during gastrulation (stage 120 in Fig. 4e) and later (Extended Data Fig. 4a). This illustrates the predictive power of our inference pipeline, which reveals new mechanical features that explain the shape of cells and embryos.

To further assess the validity of our inference method, we searched for locations in the embryo where the hypothesis of foam-like mechanical equilibrium may break down. An interesting idea is to look for junctions that are unstable for the predicted tensions. In fact, when $\gamma_{ij} > \gamma_{jk} + \gamma_{ki}$, we expect the junction $ij$ to be unstable and undergo a (possibly degenerate) T1 topological transition (Fig. 4a). Any unstable junction is therefore the sign of mechanical equilibrium breakdown that can result either (1) from a too large error in tension inference or (2) from an inadequacy of the heterogeneous foam model to describe cell arrangement or geometry[59]. In a 64-cell *P. mammillata* embryo, we found 31 unstable junctions in a total of 569 junctions (Fig. 4d). These unstable junctions are detected exclusively close to the embryo center, where the lengths of the junctions become very small, and segmentation struggles to resolve cell geometry (Fig. 4d and Extended Data Fig. 4b). In general, the percentage of unstable junctions predicted by our inference pipeline remains very low, around 3%, throughout the development of the ascidian embryo up to late neurula (Fig. 4b). This represents an even lower percentage of unstable junction length below 1% (Fig. 4c), which confirms that the tension equilibrium predicted by our inference pipeline is generally satisfied. To assess the validity of the inference, it is also useful to visualize the deviation from equilibrium using the force balance at the junctions (equation (2)). We therefore propose a visualization of the residuals $\|A_\Gamma \times \Gamma - C_\Gamma\|^2$ at each tricellular junction, as shown in Fig. 4c,d and Extended Data Fig. 4b.

To further illustrate the capabilities of our inference method, we report three aspects of early ascidian embryo development brought to light by our mechanical atlases. The early development of the ascidian is characterized by its high degree of invariance[19], and a stereotypical feature of this invariance is the bilateral symmetry of the embryo. However, each embryo shows a certain degree of geometric variability between its left and right sides, which is well reflected in the mechanical asymmetry, as illustrated by the (a)symmetry of the tension and pressure maps inferred in Fig. 5a and Extended Data Fig. 5a.

The cell fate in the ascidian embryo is also invariant, as has been described for several decades (reviewed in ref. [19]). At the 76-cell stage, the animal hemisphere is composed exclusively of ectodermal cells, while the vegetal hemisphere is segregated into neural and/or notochord progenitors and endoderm or mesoderm germ layers. We find that this patterning of cell fate is reflected in a remarkable manner in different regions of cell mechanics: ectoderm and endoderm cells have lower apical tension and lower pressure, while neural plate and mesoderm cells form very distinct regions of higher apical tension and pressure (Fig. 5b). This is probably due to the different mitotic history of each lineage, since fate specification is accompanied by an independent cell cycle timing in each specified tissue[19,57]. In the 76-cell stage, neural and/or notochord and mesoderm cells have, in fact, just undergone cell division (they are in their eighth cell cycle), while endoderm cells were born more than 40 minutes ago and are in the middle of interphase, just before they undergo apical constriction[60]. In the neurula stage, apical constriction has been reported to drive neural tube closure with greater contractility on the apical side of the nerve cord and brain tissues[61]. Consistent with this, our inference pipeline predicts on the vegetal side of the embryo at 395-, 702- and 758-cell stages a high apical tension in cells located in the anterior neural plate that are undergoing folding (Fig. 5c arrow in the vegetal pole view, Extended Data Fig. 5b). A sagittal section of the embryo at this stage reveals that the neural tube has more cortical tension than the overlying epidermis of the underlying endoderm and notochord (Fig. 5c, sagittal section); this higher tension is reflected in a stronger accumulation of myosin II in the neural tube compared to other tissues (Fig. 5c, myosin sagittal section and also ref. [61]).

Finally, we performed tension inference in the early *C. elegans* embryo from 4 to 15 cells (Fig. 6). Unlike ascidian and, to a certain extent, mouse embryos, an eggshell strongly constrains the shape of cells from the zygote stage. This confinement has shown to be an essential cue controlling early cell arrangement[62,63] and ensures Laplace's law is no longer adequate to account for cell pressures, which are directly affected by the mechanical resistance of the shell. We confirm this characteristic with 3D simulations of a four-cell embryo confined within

an ellipsoid (Fig. 6a), using realistic parameters that we previously measured in ref. 63. In this realistic simulation, we show that the mean curvature may be locally perturbed by the shell along cell–medium interfaces, especially for ABp and EMS blastomeres, which precludes the use of Laplace's law that assumes constant mean curvature interfaces. Indeed, when we infer pressures with the Laplace or Laplace variational formula on this mesh, we obtain pressure predictions, which are 20 to 30% different from the actual value in the four blastomeres (Fig. 6a). Therefore, simultaneous tension and pressure inference may not be a good strategy in this case[33], while breaking down the inference in two successive steps still allows us to infer tensions independently of cell pressures. We find, in agreement with the measurements in ref. 63, a lower cell–medium tensions in P2 and EMS cells in the four-cell stage *C. elegans* embryo, and predict a general trend of lower cell–medium cortical tension in descendants of the P-lineage at subsequent stages of embryo development (Fig. 6b).

## Discussion

We presented a robust end-to-end computational pipeline to infer relative surface tensions and pressures directly from 3D fluorescent images of embryos or tissues. It is based, in particular, on a new and fast method for generating surface meshes from cell segmentation masks, which allows for a more accurate extraction of geometric features than previous approaches[46]. Therefore, our algorithm is compatible with the latest segmentation methods[47,64,65] and can scale to thousands of cells. We also introduced a new formula for inferring pressures from a triangle surface mesh, which outperforms the direct inversion of Laplace's law. By performing a systematic sensitivity analysis on simulated embryos, we showed that the classic Young–Dupré formula gives the best tension inference results for moderate noise in the image or in the cell shape. Our pipeline intrinsically achieves maximum relative force errors of roughly 10% from images of simulated embryos (Fig. 3e–f and Supplementary Notes C.4.2 and C.5.3). Additionally, we provide several visualization tools built on Polyscope[66] to display multicellular morphology and forces in multiple ways, including a force graph representation of the cell aggregate and a 3D map of cellular stress tensors (Fig. 2h–i). The residues and predicted topological changes of inference for each junction in the aggregate can also be directly plotted to enable local evaluation of the method and/or the active foam hypothesis (Fig. 4b,c). Subsequently, we demonstrated the biological relevance of our approach by generating mechanical atlases of the early ascidian embryo: our inference method can recover characteristic patterns of apical tension previously observed[61], including a lower apical tension measured in mitotic cells before the 64-cell stage[6]. It can also make new predictions and reveal mirroring patterns of cell mechanics and cell fate in germ layers. Finally, we demonstrate the use of decoupling pressure and tension inference by applying our methodology to the early *C. elegans* embryo, which develops within a shell.

One forthcoming challenge will be to generate spatiotemporal mechanical atlases of various embryos. Indeed, a temporal reference is so far missing to calibrate the successive spatial maps in time. As demonstrated in two dimensions[63], combining static inference with the temporal measurement of absolute forces in a single location, or imaging phosphomyosin fluorescence intensity as a proxy for tension, could become a generic approach to construct temporal atlases of absolute mechanical forces, but this needs to be repeated in three dimensions.

A second challenge will involve the inclusion of junctional mechanics in the form of additional line tension contributions at the apical surface of cells. Indeed, blastomeres with a contact to the cell medium acquire generally apico-basal polarity short before the blastula stage in early embryos. This emergence of apical polarity is generally associated with the formation of tight junctions and a contractile ring of actomyosin delimiting each apical surface[67], that is expected to create additional line tensions at tricellular junctions. The question of the uniqueness of the inverse solution will furthermore arise, since several stable discontinuous bifurcation states can exist in the presence of line and surface tensions[68], which will first require a in-depth theoretical effort.

A third challenge will consist of generalizing force-inference methods to more complex mechanical models, such as recent active viscous surface models, which naturally generate inhomogeneous and anisotropic surface tensions[69], as well as possible torques, leading to more complex shapes and force balance equations. This will be particularly important for precisely characterizing the mechanics of dividing cells[43,68] and faster growing organisms, such as *C. elegans*, for which the time scales of visco-active relaxation and development may no longer be well separated. A possible generic avenue to solve these problems may lie in a fully variational approach, where a mathematical loss between the microscopy images and the meshes could be constrained by an arbitrary mechanical model to allow direct gradient-based optimization of its spatiotemporal parameters. Our recent effort to design such an efficient loss for comparing a mesh and an image may begin to fill this gap[70]. The current force-inference method we introduced will remain a fundamental building block to this research field, providing already accurate geometric and mechanical maps, which will form an ideal initial guess to refined but more computationally expensive iterative methods.

With a documented and user-friendly implementation in Python, our 3D force-inference method can be easily applied to 3D images of embryos or small tissues undergoing a sufficiently slow development, and can be combined with spatial 'omic' data generated in early embryos to uncover possible mechanochemical couplings. 3D force-inference complements the growing range of tools available for studying the mechanical properties of tissues in space and time[2], and we anticipate that this approach will help explain the mechanical underpinnings of large-scale morphogenetic movements at the cellular level and illuminate the intricate interplay between chemical signaling and mechanics during development. By revealing the developmental forces shaping organisms, our method may open new evo–devo studies, such as the investigation of the mechanical differences between closely related phylogenetic neighbors or the understanding of the mechanical aspects contributing to the divergence of developmental pathways in evolution.

## Online content

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

## Methods

### Variants of Young–Dupré formulas

Starting from the vectorial expression of the Young–Dupré law (equation (2)) we call its decomposition simply by Young–Dupré its decomposition with cosines of polar angles:

$$\gamma_{ij} + \gamma_{jk} \cos \alpha_{ki} + \gamma_{ki} \cos \alpha_{jk} = 0$$

$$\gamma_{ij} \cos \alpha_{ki} + \gamma_{jk} + \gamma_{ki} \cos \alpha_{ij} = 0$$

$$\gamma_{ij} \cos \alpha_{jk} + \gamma_{jk} \cos \alpha_{ij} + \gamma_{ki} = 0$$

Another set involves both cosines and sines of angles made by vectorial tensions with one direction chosen arbitrarily choose along a tension vector, and we call it the Young–Dupré projection:

$$\gamma_{ij} + \gamma_{jk} \cos \alpha_{ki} + \gamma_{ki} \cos \alpha_{jk} = 0$$

$$\gamma_{jk} \sin \alpha_{ki} - \gamma_{ki} \sin \alpha_{jk} = 0$$

Many other mathematically equivalent formulas may in fact be derived from trigonometric laws applied to the triangle formed by vectorial tensions (Supplementary Note). Here, we will also use Lami's theorem, which derives directly from the law of sines and was proposed as an alternative formula for tension inference in two dimensions[34,56]:

$$\frac{\gamma_{ij}}{\sin \alpha_{ij}} = \frac{\gamma_{jk}}{\sin \alpha_{jk}} = \frac{\gamma_{ki}}{\sin \alpha_{ki}} \tag{3}$$

To avoid divergence at small polar angles, it was proposed to consider the same equations written as $\gamma_{ij} \sin \alpha_{jk} = \gamma_{jk} \sin \alpha_{ij}$, $\gamma_{jk} \sin \alpha_{ki} = \gamma_{ki} \sin \alpha_{jk}$, which we call inverse Lami, or to consider the logarithm of the equation (3), that we call the Lami logarithm.

### Biological material

The eggs of the ascidian *P. mammillata* were harvested from animals obtained in Sète and kept in the laboratory in a tank of natural seawater at 16 °C. Egg preparation and microinjection have been previously described (detailed protocols in refs. 72,73). Eggs and sperm were collected by dissection. Sperm was activated in pH 9.0 seawater before fertilization (see the detailed protocol in ref. 73). All imaging experiments were performed at 20 °C.

### Plasma membrane and Myosin II fluorescent labeling

The plasma membrane was imaged using our characterized construct PH::Tomato[73] whereas Myosin II was imaged using Myosin II intrabody iMyo (called SF9::GFP in Chaigne et al.[74] the plasmid pRN3-SF9-GFP is a kind gift from the M.H. Verlhac laboratory). RNAs coding for PH::Tomato (1 µg µl⁻¹) and SF9/iMyo::GFP (4 µg µl⁻¹) were injected in unfertilized Phallusia oocytes that were then fertilized between 2 and 12 h after injection.

### Confocal imaging of *P. mammillata* embryos

Four-dimensional confocal imaging was performed at 20 °C using a Leica TCS SP8 inverted microscope equipped with hybrid detectors and a ×20/0.8 numerical aperture water objective lens. A 3D stack was taken every minute with a pixel size of $1 \times 1$ µm and a $z$ step of 1 µm (to obtain cubic voxels). The *Phallusia* embryos shown in Fig. 4d (and Extended Data Fig. 4) from 16 cells to 32 cells were imaged in the Team ABC laboratory, while embryos from stage 64 cells and later stages (shown in Figs. 1, 4 and 5 and Extended Data Figs. 4 and 5) were obtained from a public dataset of segmented *P. mammillata* embryos published in ref. 19.

### Statistics and reproducibility

The boxplots (shown in Fig. 3e,f and Extended Data Fig. 3b) are realized with the default parameters of the boxplot function of the matplotlib Python library. The box center is located at the median, and its

extremities represents the first and third quartiles. The whiskers are located at Q1 − 1.5× (Q3 − Q1) and Q3 + 1.5× (Q3 − Q1) where Q denotes the quartiles.

The shaded regions in plots show the standard deviation (in Figs. 3b,c and 2b and Extended Data Fig. 2b). The fluorescence microscopy images of Myosin II shown on Fig. 5 are representative of $n = 4$ experiments.

### Reporting summary

Further information on research design is available in the Nature Portfolio Reporting Summary linked to this article.

## Data availability

Images and segmentation masks are already available publicly for *P. mammillata* embryos on figshare (greater than or equal to 64 cells)[19] and for *C. elegans* embryos on figshare[18]. The simulated dataset (original simulation meshes, artificial images, segmentation masks and tensions/pressures) used to benchmark the method is available publicly on Zenodo[54]. Additional experimental images of ascidian embryos (fewer than 64 cells) and their segmentation masks are available upon request.

## Code availability

Our inference pipeline foambryo[39] is distributed as a standalone Python package in the repository PyPI (https://pypi.org/project/foambryo/) and the source code is available on GitHub (https://github.com/VirtualEmbryo/foambryo) and is archived on Zenodo[39]. The companion code Delaunay-watershed[55] is installed automatically with foambryo, but is also distributed as a standalone Python package in the repository PyPI and the source code available on GitHub. The mesh reconstruction pipeline Delaunay-watershed[55] is also distributed as a separate Python package on PyPI (https://pypi.org/project/delaunay-watershed-3d/) and its source code is on GitHub (https://github.com/VirtualEmbryo/delaunay-watershed) and archived on Zenodo[55].

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

## Acknowledgements

This project has received funding from the European Research Council under the Horizon 2020 research and innovation program of the European Union (grant agreement no. 949267 to H.T.). S.I. was funded by Ecole Polytechnique (AMX grant). H.T. has been supported by EMBRC-France (AAP Découverte 2020), by the Bettencourt-Schueller Foundation, by the CNRS and the Collège de France. R.D. and A.M. are supported by a grant from the French Government funding agency Agence Nationale de la Recherche to A.M. (grant no. ANR 'MorCell': ANR-17-CE13-0028) and received funding from the MITI at CNRS (AAP Modélisation du vivant). We are grateful for continuous support to the Imaging Platform and the Animal Facility of the Institut de la Mer de Villefranche, which is supported by EMBRC-France, whose French state funds are managed by the ANR within the Investments of the Future program under reference no. ANR-10-INBS-0. We thank J.-L. Maitre for sharing microscopy images of eight-cell-stage mouse embryos, and F. Graner as well as all members of the Turlier team for discussions.

## Author contributions

H.T. supervised the project and acquired funding. S.I., F.D. and H.T. developed the theory. S.I. designed the computational pipeline

and performed all computations. R.D. and A.M. performed the experiments. S.I., R.D. and H.T. analyzed the data. S.I. made the figures with input from H.T. H.T. wrote the manuscript with the input of all authors.

## Competing interests

The authors declare no competing interests.

## Ethics declarations

This research complies with all relevant ethical regulations.

## Additional information

**Extended data** is available for this paper at https://doi.org/10.1038/s41592-023-02084-7.

**Correspondence and requests for materials** should be addressed to Hervé Turlier.

**a**

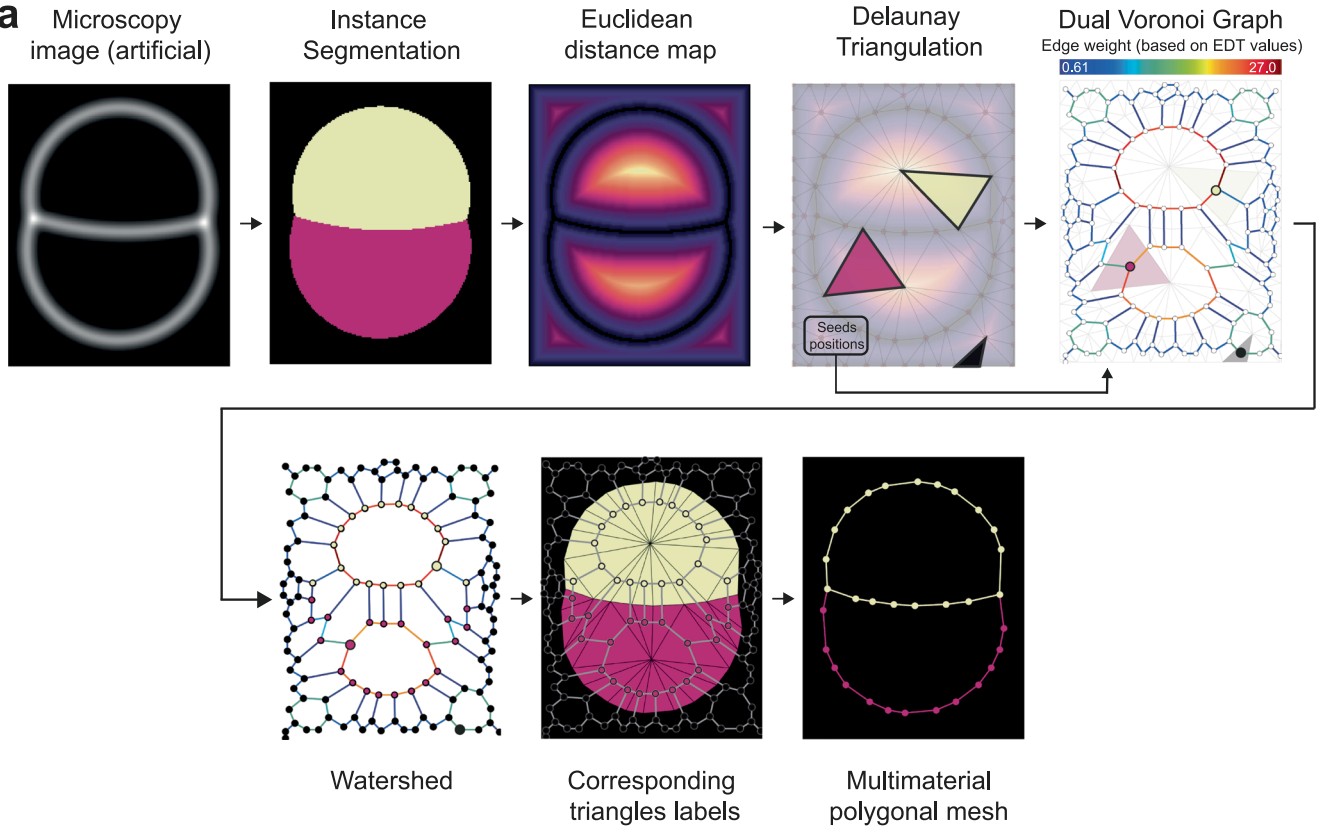

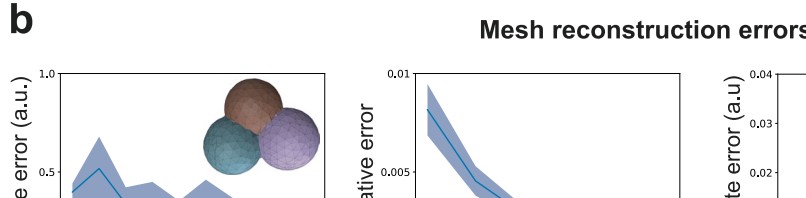

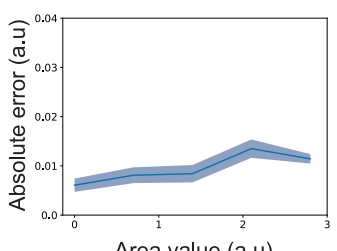

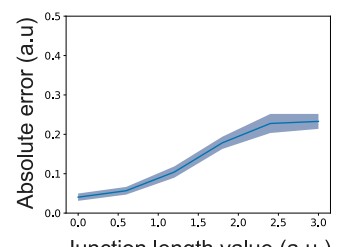

**b**

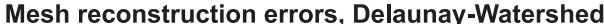

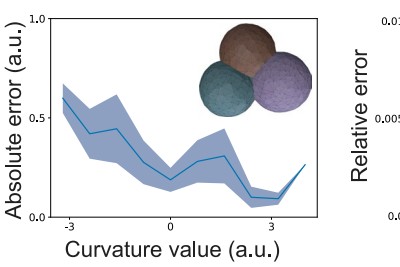

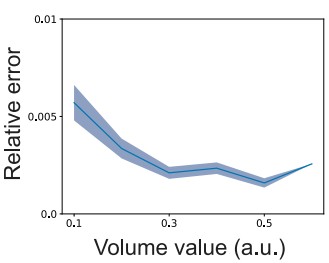

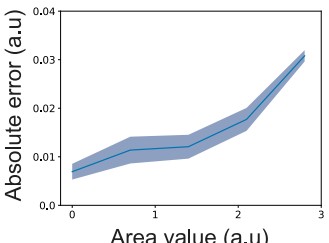

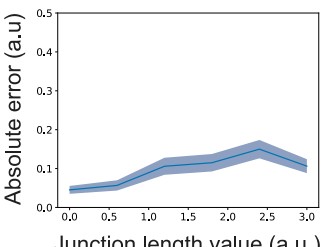

**Extended Data Fig. 1 | Detailed procedure and benchmarking of the Delaunay-watershed mesh generation algorithm. a)** Pipeline for mesh generation from a microscopy image (here in 2D for graphical purposes). From the Delaunay triangulation of the image domain, we construct a graph of the dual Voronoi diagram. The edge weights of this graph are computed by integrating the value of the Euclidean distance map along corresponding edges that separates two triangles in the primary domain. The watershed is performed on the dual graph, and the seeds are chosen by taking the triangles containing the pixel with the highest EDT value in the primary domain. **b)** Comparison of the geometric error (mean ± SD) obtained on interface curvatures, cell volumes, interface areas, and junctional lengths between CGAL and our Delaunay-watershed algorithms for mesh reconstruction (n=54 simulated embryos).

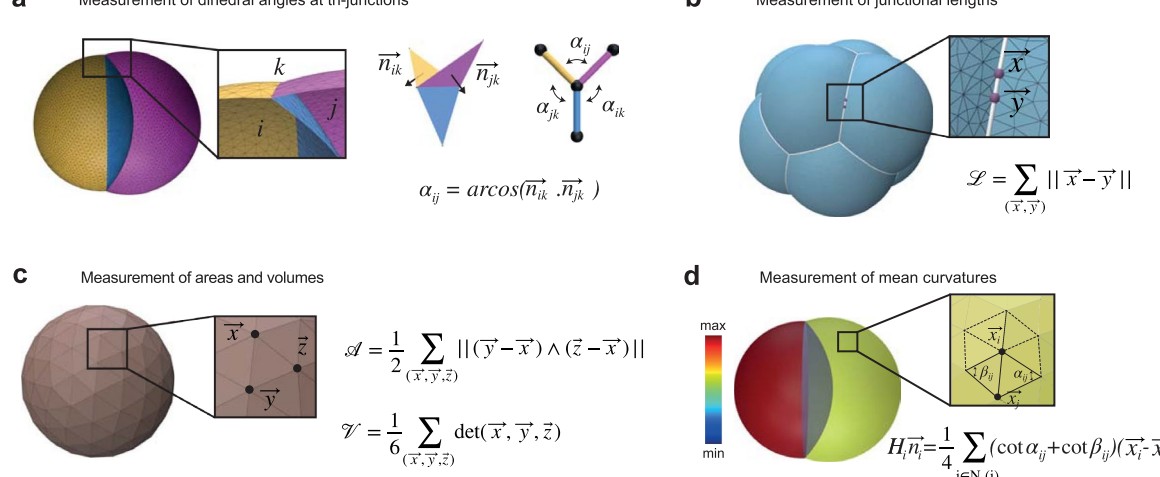

**a** Measurement of dihedral angles at tri-junctions

$$\alpha_{ij} = arcos(\overrightarrow{n_{ik}} . \overrightarrow{n_{jk}})$$

**b** Measurement of junctional lengths

$$\mathscr{L} = \sum_{(\vec{x}, \vec{y})} || \vec{x} - \vec{y} ||$$

**c** Measurement of areas and volumes

$$\mathscr{A} = \frac{1}{2} \sum_{(\vec{x}, \vec{y}, \vec{z})} || (\vec{y} - \vec{x}) \wedge (\vec{z} - \vec{x}) ||$$

$$\mathscr{V} = \frac{1}{6} \sum_{(\vec{x}, \vec{y}, \vec{z})} det(\vec{x}, \vec{y}, \vec{z})$$

**d** Measurement of mean curvatures

$$H_i \vec{n_i} = \frac{1}{4} \sum_{j \in N_1(i)} (\cot\alpha_{ij} + \cot\beta_{ij})(\overrightarrow{x_i} - \overrightarrow{x_j})$$

**Extended Data Fig. 2 | Measurement of geometrical quantities on nonmanifold multimaterial triangle surface meshes.** a) Contact angles are calculated at each junction as the mean of dihedral angles in each triplet of triangles that constitutes the junction. A dihedral angle is computed from the unit normals to the two adjacent triangles. b) Junctions are lines that separate three different materials or regions (three cells or 2 cells and the cell medium).

Their length can be easily defined and measured with our nonmanifold mesh data structure. c) Each cell is represented by a bounded volume (a discrete manifold). We can compute their volumes and areas from our multimaterial mesh data structure with formulas derived in the Supplementary Note. d) Mean discrete curvatures can be computed using the cotangent formula (see Supplementary Note).

**a**

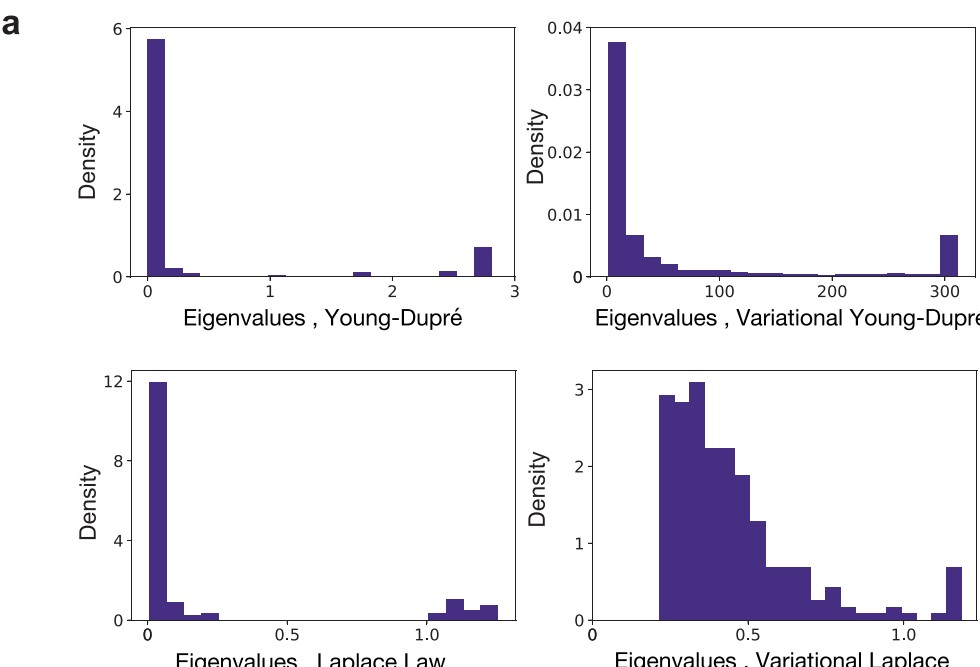

**b** Pressure inference errors

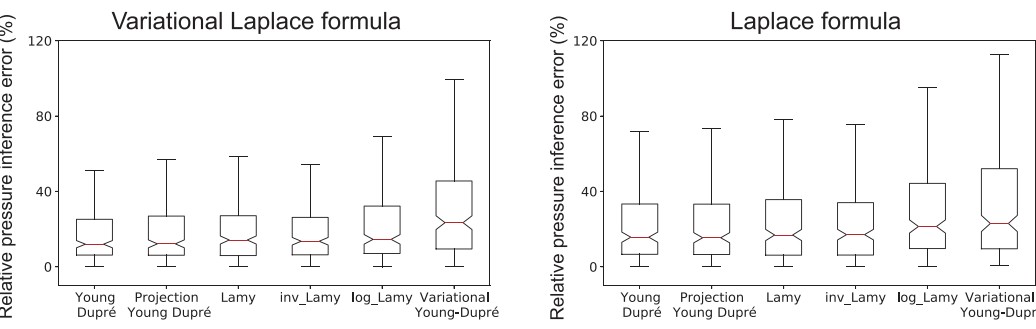

**c** Inference error dependence on image size and control-point distance

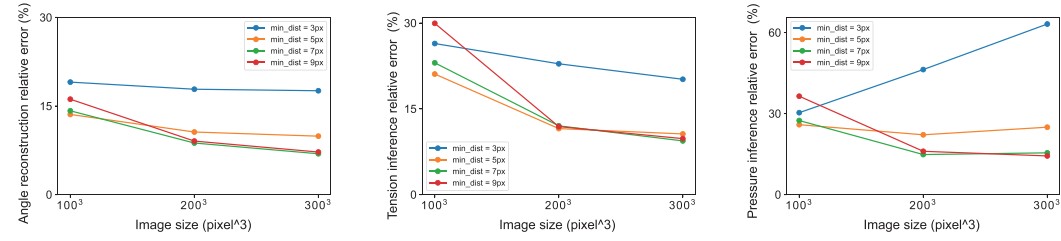

**d** Comparison of manually and automatically inferred tensions variability in the 8-cell mouse embryo

Variability in tensions inferred automatically with our pipeline

| | Embryo 1 | Embryo 2 | Embryo 3 | Embryo 4 |
|---|---|---|---|---|
| $\mathrm{std}(\gamma_{cm})/\mathrm{mean}(\gamma_{cm})$ | 0.26 | 0.27 | 0.18 | 0.24 |
| $\mathrm{std}(\gamma_{cc})/\mathrm{mean}(\gamma_{cc})$ | 0.10 | 0.05 | 0.08 | 0.07 |

Variability in tensions inferred manually (data from ref. [5])

| Time post-division (hh:mm) | 0:00 | 3:00 | 6:00 | 9:00 | 12:00 |
|---|---|---|---|---|---|
| $\mathrm{std}(\gamma_{cm})/\mathrm{mean}(\gamma_{cm})$ | 0.27 | 0.27 | 0.29 | 0.24 | 0.31 |
| $\mathrm{std}(\gamma_{cc})/\mathrm{mean}(\gamma_{cc})$ | 0.25 | 0.26 | 0.48 | 0.41 | 0.43 |

**Extended Data Fig. 3 | See next page for caption.**

**Extended Data Fig. 3 | Inference sensitivity and influence of tensions formulas for pressure inference.** a) Histogram of eigenvalues of the pseudo-inverse matrices used to infer tensions and pres- sures for the Young–Dupré, variational Young-Dupré, Laplace and Variational Laplace formulas, on our simulated embryo dataset. The spread of the histogram is a measure of the conditioning of the matrix. b) Comparison of the relative error on inferred pressures obtained on our simulated embryo dataset between Laplace and variational Laplace formulas (n=54 simulated embryos; the red center line denotes the median value, while the box contains the 25th to 75th percentiles of dataset; the whiskers mark the 5th and 95th percentiles). c) Mean relative error on angles reconstruction (left), tension inference (mid- dle) and pressure inference (right), depending on the refinement of the mesh (in pixels) and the image size. d) Comparison of the variability (SD/mean) in tensions measured manually (from[5]) and inferred automatically with our pipeline in the 8-cell mouse.

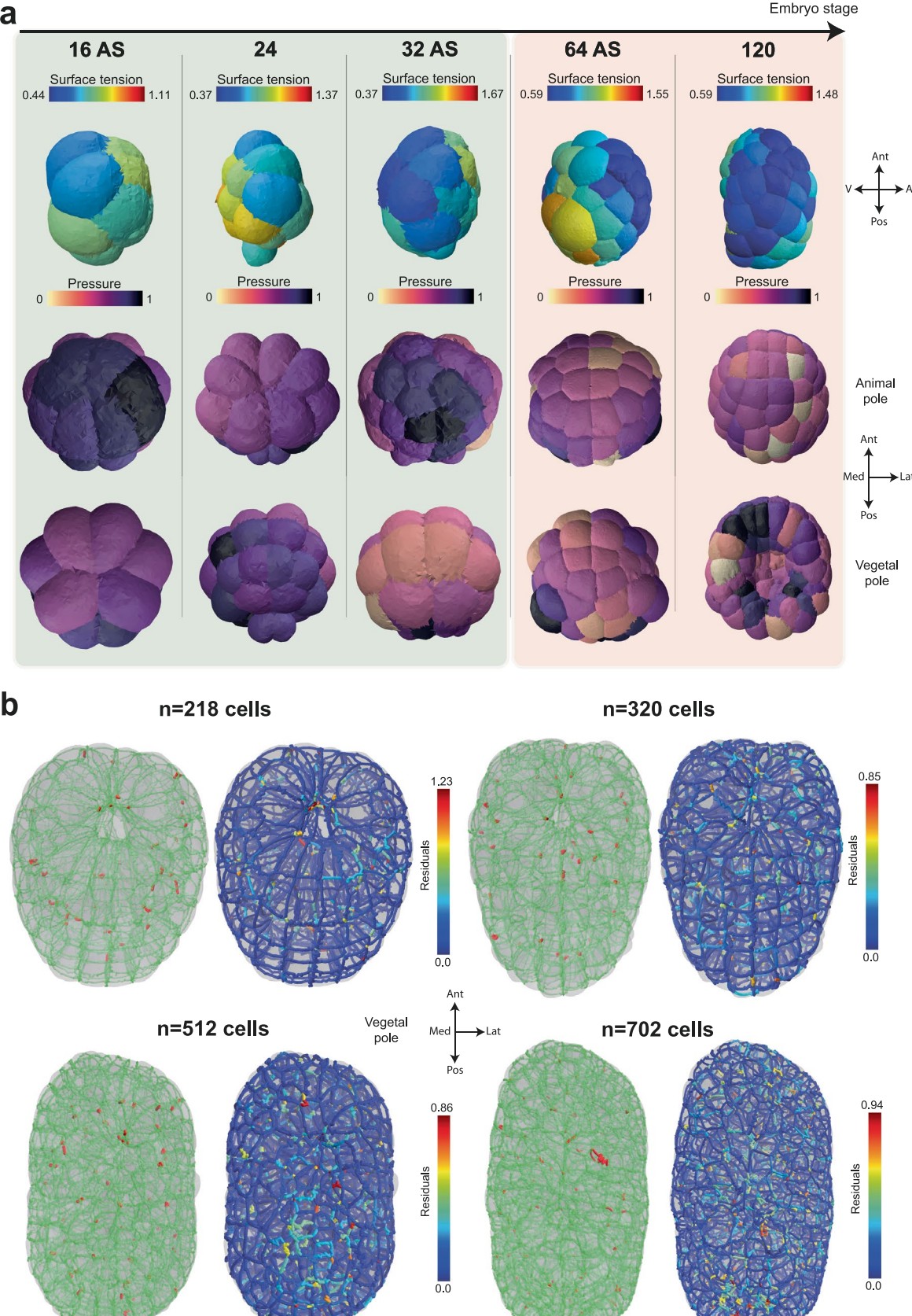

**Extended Data Fig. 4 | Additional validation data of the 3D tension inference.**
a) Mitotic softening and stiffening in the 16AS, 24, 32AS, 64 and 120 cell stages of
the early ascidian embryo (P. mammillata). Upper row: sagittal view of inferred
apical tension. Middle and bottow rows: animal and vegetal views of the inferred
cell pressures. The ratio of mitotic to interphase apical tension is colored green if
it is less than 1 and red if it is greater than 1. The orientation of the embryo is given
by arrows Ant: anterior, Pos: posterior, Med: medial, Lat: lateral, V: vegetal, A:
animal. b) Vegetal view of stable (green) and unstable (red) junctions (Left) and
tension inference 512 and 702 cell stages. The orientation of the embryo is given
by arrows Ant: anterior, Pos: posterior, Med: medial, Lat: lateral.

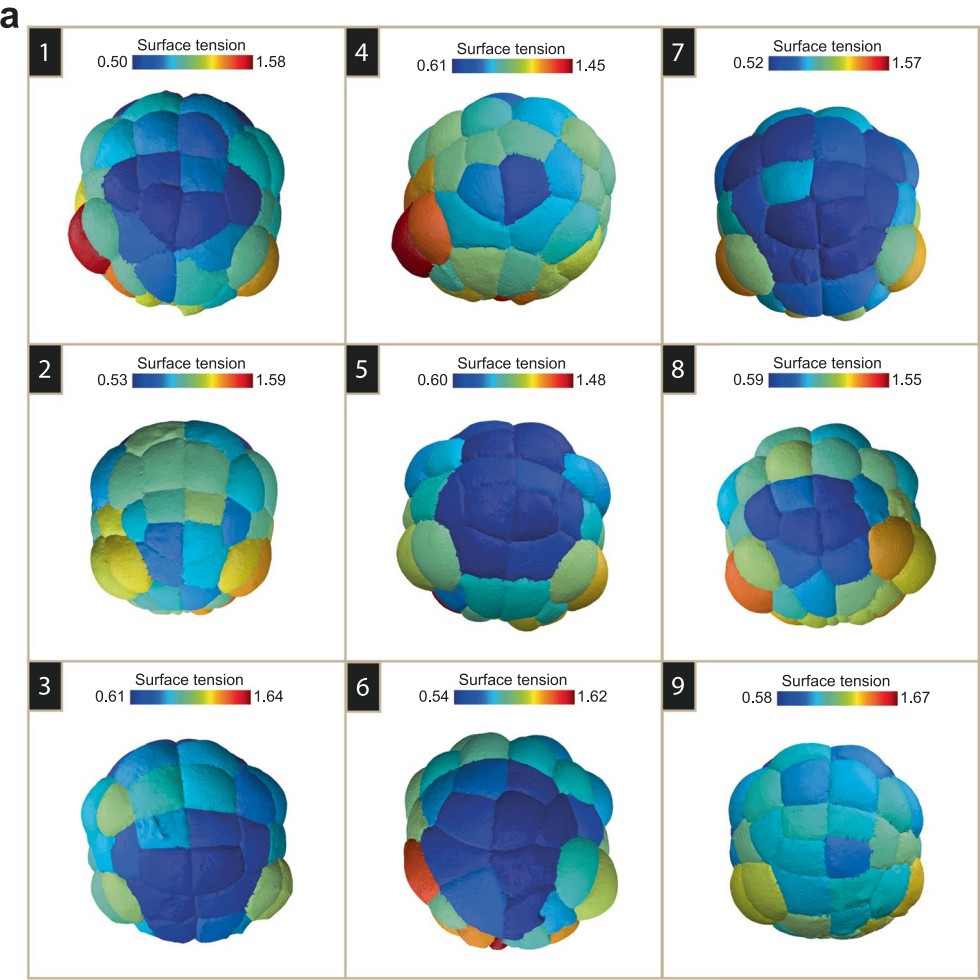

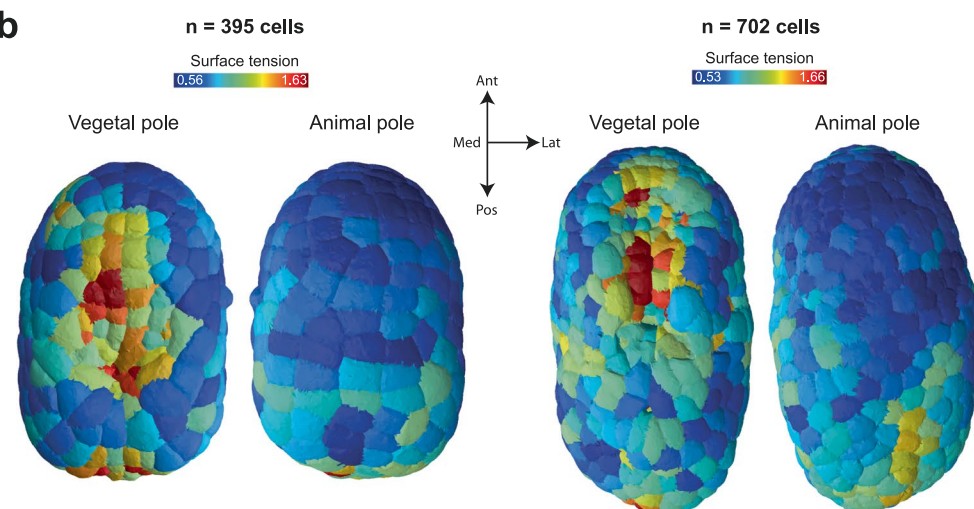

**Extended Data Fig. 5 | Additional tension maps of ascidian P. mammillata gastrula and neurula.** a) Nine examples of apical tension maps of 64 cell gastrula (animal pole). b) Maps of apical tensions at the animal and vegetal poles of early (Left) and late neurula ascidian embryos. The orientation of the embryo is given by arrows Ant: anterior.

# Reporting Summary

## Statistics

For all statistical analyses, confirm that the following items are present in the figure legend, table legend, main text, or Methods section.

| n/a | Confirmed | |
|---|---|---|
| ☐ | ☒ | The exact sample size (*n*) for each experimental group/condition, given as a discrete number and unit of measurement |
| ☒ | ☐ | A statement on whether measurements were taken from distinct samples or whether the same sample was measured repeatedly |
| ☒ | ☐ | The statistical test(s) used AND whether they are one- or two-sided<br>*Only common tests should be described solely by name; describe more complex techniques in the Methods section.* |
| ☒ | ☐ | A description of all covariates tested |
| ☒ | ☐ | A description of any assumptions or corrections, such as tests of normality and adjustment for multiple comparisons |
| ☐ | ☒ | A full description of the statistical parameters including central tendency (e.g. means) or other basic estimates (e.g. regression coefficient) AND variation (e.g. standard deviation) or associated estimates of uncertainty (e.g. confidence intervals) |
| ☒ | ☐ | For null hypothesis testing, the test statistic (e.g. *F*, *t*, *r*) with confidence intervals, effect sizes, degrees of freedom and *P* value noted<br>*Give P values as exact values whenever suitable.* |
| ☒ | ☐ | For Bayesian analysis, information on the choice of priors and Markov chain Monte Carlo settings |
| ☒ | ☐ | For hierarchical and complex designs, identification of the appropriate level for tests and full reporting of outcomes |
| ☒ | ☐ | Estimates of effect sizes (e.g. Cohen's *d*, Pearson's *r*), indicating how they were calculated |

*Our web collection on statistics for biologists contains articles on many of the points above.*

## Software and code

Policy information about availability of computer code

| Data collection | No specific code was used to collect data. |
|---|---|
| Data analysis | Python >3.6 and classical associated scientific libraries (NumPy >1.25, SciPy >1.11, NetworkX >3.1, Matplotlib, scikit-image >1.0) were used for data analysis. Polyscope >1.3.0 (https://polyscope.run) was used to build a custom viewer for visualizing inference results on reconstructed meshes. CGAL 5.5.1 (https://www.cgal.org) was used to compare our mesh reconstruction with state-of-the-art techniques from computer graphics.<br>We developed two custom codes, that will be available on our GitHub repository and on the Python package index PyPi (https://pypi.org) upon publication:<br>- the Delaunay-watershed code for mesh reconstruction from segmentation masks: https://github.com/VirtualEmbryo/delaunay-watershed<br>- the foambryo code for 3D force inference and visualization: https://github.com/VirtualEmbryo/foambryo |

For manuscripts utilizing custom algorithms or software that are central to the research but not yet described in published literature, software must be made available to editors and reviewers. We strongly encourage code deposition in a community repository (e.g. GitHub). See the Nature Portfolio guidelines for submitting code & software for further information.

## Data

Policy information about <u>availability of data</u>

All manuscripts must include a <u>data availability statement</u>. This statement should provide the following information, where applicable:

- Accession codes, unique identifiers, or web links for publicly available datasets
- A description of any restrictions on data availability
- For clinical datasets or third party data, please ensure that the statement adheres to our <u>policy</u>

> Images and segmentation masks are already available publicly for P. Mammillata embryos on figshare (≥ 64 cells) (https://figshare.com/projects/ Phallusia_mammillata_embryonic_development/64301) and for C. elegans embryos on figshare (https://figshare.com/articles/journal_contribution/ CShaper_Supplementary_Data/12839315). The simulated dataset (artificial images, segmentation masks and tensions/pressures) used to benchmark the method is available publicly on Zenodo (https://zenodo.org/record/7881017). Additional experimental images of ascidian embryos ($<$ 64 cells) and their segmentation masks are available upon request.

## Human research participants

Policy information about <u>studies involving human research participants and Sex and Gender in Research.</u>

| | |
|---|---|
| Reporting on sex and gender | N/A |
| Population characteristics | N/A |
| Recruitment | N/A |
| Ethics oversight | N/A |

Note that full information on the approval of the study protocol must also be provided in the manuscript.

# Field-specific reporting

Please select the one below that is the best fit for your research. If you are not sure, read the appropriate sections before making your selection.

☒ Life sciences          ☐ Behavioural & social sciences          ☐ Ecological, evolutionary & environmental sciences

For a reference copy of the document with all sections, see nature.com/documents/nr-reporting-summary-flat.pdf

# Life sciences study design

All studies must disclose on these points even when the disclosure is negative.

| | |
|---|---|
| Sample size | Figures 4, 5 and 6b: among n>3 samples we provided the most representative microscopy image or mechanical pattern. No averaging of mechanical maps was performed; the comparisons of myosin and surface tension patterns (Figure 5c) are shown for illustrative purpose. |
| Data exclusions | We did not exclude data: for instance in Extended data Figure 5a, we present most of the embryos available in the public repository. For other figures, we selected representative mechanical maps inferred from microscopy images based on their relevance to published data or emerging patterns. |
| Replication | We found quantiatively consistent replicates for the mechanical patterns observed in Figures 4e (n>3), Figure 5 and Figure 6b |
| Randomization | We did not allocate samples to different groups, but retained all imaged embryos displaying a normal development. |
| Blinding | N/A |

# Reporting for specific materials, systems and methods

We require information from authors about some types of materials, experimental systems and methods used in many studies. Here, indicate whether each material, system or method listed is relevant to your study. If you are not sure if a list item applies to your research, read the appropriate section before selecting a response.

## Materials & experimental systems

| n/a | Involved in the study |
|-----|----------------------|
| ☒ ☐ | Antibodies |
| ☒ ☐ | Eukaryotic cell lines |
| ☒ ☐ | Palaeontology and archaeology |
| ☐ ☒ | Animals and other organisms |
| ☒ ☐ | Clinical data |
| ☒ ☐ | Dual use research of concern |

## Methods

| n/a | Involved in the study |
|-----|----------------------|
| ☒ ☐ | ChIP-seq |
| ☒ ☐ | Flow cytometry |
| ☒ ☐ | MRI-based neuroimaging |

# Animals and other research organisms

Policy information about studies involving animals; ARRIVE guidelines recommended for reporting animal research, and Sex and Gender in Research

| Laboratory animals | The study did not involve laboratory animals |
|--------------------|----------------------------------------------|
| Wild animals | Adults from the ascidian specie Phallusia mammillata were collected in Sète by scuba divers and transported to the laboratory in Villefranche-sur-mer in a tank of seawater at 16°C by car (371km). Animals are dissected to collect their eggs and sperm. |
| Reporting on sex | N/A |
| Field-collected samples | The eggs of the ascidian Phallusia mammillata were harvested from animals and kept in the laboratory in a tank of natural seawater at 16°C. |
| Ethics oversight | No ethical approval or guidance required. All protocols are published: https://doi.org/10.1007/978-1-61779-210-6_14 |

Note that full information on the approval of the study protocol must also be provided in the manuscript.

