## [Peer Review File · Nature Methods]

Peer Review Information

Manuscript Title: Embryo mechanics cartography: inference of 3D force atlases from fluorescence microscopy

Corresponding author name(s): Hervé Turlier

Editorial Notes: n/a

Reviewer Comments & Decisions:

Decision Letter, initial version:

Dear Hervé,

Your Article, "Embryo mechanics cartography: inference of 3D force atlases from fluorescence microscopy", has now been seen by 3 reviewers. As you will see from their comments below, although the reviewers find your work of considerable potential interest, they have a few suggestions for improvement. We are interested in the possibility of publishing your paper in Nature Methods, but would like to consider your response to these concerns before we reach a final decision on publication.

We therefore invite you to revise your manuscript to address these concerns. In particular, please make sure that the code does not have errors and can be tested by the reviewers.

* include a point-by-point response to the reviewers and to any editorial suggestions

* please underline/highlight any additions to the text or areas with other significant changes to facilitate review of the revised manuscript

- * address the points listed described below to conform to our open science requirements
- * ensure it complies with our general format requirements as set out in our guide to authors at www.nature.com/naturemethods
- * resubmit all the necessary files electronically by using the link below to access your home page

[Redacted] This URL links to your confidential home page and associated information about manuscripts you may have submitted, or that you are reviewing for us. If you wish to forward this email to co-authors, please delete the link to your homepage.

We hope to receive your revised paper within four weeks. If you cannot send it within this time, please let us know. In this event, we will still be happy to reconsider your paper at a later date so long as nothing similar has been accepted for publication at Nature Methods or published elsewhere.

OPEN SCIENCE REQUIREMENTS

REPORTING SUMMARY AND EDITORIAL POLICY CHECKLISTS

Please note that these forms are dynamic 'smart pdfs' and must therefore be downloaded and completed in Adobe Reader. We will then flatten them for ease of use by the reviewers. If you would

like to reference the guidance text as you complete the template, please access these flattened versions at <http://www.nature.com/authors/policies/availability.html>.

DATA AVAILABILITY

We strongly encourage you to deposit all new data associated with the paper in a persistent repository where they can be freely and enduringly accessed. We recommend submitting the data to discipline-specific and community-recognized repositories; a list of repositories is provided here:

<http://www.nature.com/sdata/policies/repositories>

All novel DNA and RNA sequencing data, protein sequences, genetic polymorphisms, linked genotype and phenotype data, gene expression data, macromolecular structures, and proteomics data must be deposited in a publicly accessible database, and accession codes and associated hyperlinks must be provided in the “Data Availability” section.

Please include a “Data availability” subsection in the Online Methods. This section should inform readers about the availability of the data used to support the conclusions of your study, including accession codes to public repositories, references to source data that may be published alongside the paper, unique identifiers such as URLs to data repository entries, or data set DOIs, and any other statement about data availability. At a minimum, you should include the following statement: “The data that support the findings of this study are available from the corresponding author upon request”, describing which data is available upon request and mentioning any restrictions on availability. If DOIs are provided, please include these in the Reference list (authors, title, publisher (repository name), identifier, year). For more guidance on how to write this section please see: <http://www.nature.com/authors/policies/data/data-availability-statements-data-citations.pdf>

CODE AVAILABILITY

Please include a “Code Availability” subsection in the Online Methods which details how your custom code is made available. Only in rare cases (where code is not central to the main conclusions of the paper) is the statement “available upon request” allowed (and reasons should be specified).

MATERIALS AVAILABILITY

SUPPLEMENTARY PROTOCOL

To help facilitate reproducibility and uptake of your method, we ask you to prepare a step-by-step Supplementary Protocol for the method described in this paper. We [encourage authors to share their step-by-step experimental protocols](https://www.nature.com/nature-research/editorial-policies/reporting-standards#protocols) on a protocol sharing platform of their choice and report the protocol DOI in the reference list. Nature Portfolio's Protocol Exchange is a free-to-use and open resource for protocols; protocols deposited in Protocol Exchange are citable and can be linked from the published article. More details can found at www.nature.com/protocolexchange/about.

ORCID

Sincerely,
Madhura

Madhura Mukhopadhyay, PhD
Senior Editor
Nature Methods

Reviewers' Comments:

Reviewer #1:

Remarks to the Author:

The present manuscript presents an analysis pipeline and algorithm for force-inference measurements in 3D. It includes an advanced and superior mesh reconstruction method and a robust inversion algorithm. the method is tested on active foam simulations and on data of early embryos of mice, c. elegant and ascidian. Importantly, all in 3D, which is far from trivial.

the article is clearly written, and the obtained precision of analysis is pretty impressive. the proper use of force inference technology is partly hampered by the lack of such beautiful paper and code. I am convinced that this manuscript will have a huge impact in the field and helps to perform more quantitative analysis in 3D cellular systems.

Reviewer #2:

Remarks to the Author:

This manuscript represents a breakthrough. The work it describes is serious, systematic, validated and clearly explained. After suitable revision and polishing, it will be instrumental in making 3D stress inference a mature technique of great interest to investigate living tissue stresses in situ, in a non invasive way, and with a high throughput.

* Vocabulary :

- Keep "surface tension" for the interface between two mediums or tissues (eg the whole embryo "approximated to a droplet"). Here use "cell-cell tension", "cell-cell junction tension", "junction tension" or any other precise term.

- "Pressure inference" and "Tension inference" are correct terms. But "Force inference", or "inference" alone, should be replaced with "stress inference" (see Noll 2020), "cell stress inference", "tension and pressure inference" any other precise term.

- The term "active foam", applied to a living tissue, underlines the analogy and differences between a foam and a tissue. It has been popularized in a context where it was correct : a discussion of the effect of cell ATP consumption on the analogy validity. Here the difference between the tissue and the foam lies in the fact that tension vary from one cell junction to another. The effects of this spatial heterogeneity is completely independent of its origin (whether active or not) and the current manuscript can be applied to passive heterogeneous foams too. Hence replace "active foam", which is ambiguous, with "heterogeneous foam" or any other precise term.

- The Surface Evolver is not a "vertex model".

- In addition, The Surface Evolve can be called a simulation "on a discrete mesh" but this can create confusion with simulations on a discrete lattice. Preferably call it a simulation "of discretised junctions", "of triangulated junctions" or any other precise term.

* Other :

In introduction, discuss Noll's approach to simultaneously segment the pattern and equilibrate it in 2D ; then Liu, Lemaire et al's preprint attempting at doing the same in 3D, and its limitations ; and then finally how the current manuscript is positioned. The exact approach should be better explained, underlining what is standard, what is new, the number of cells which can be treated, and what validations are performed.

Reviewer #3:

Remarks to the Author:

The authors presented a computational framework that uses microscopic images of tissues to generate 3D meshes, which are then used to estimate surface tensions and cell pressures. The computational framework was tested on a few simple artificial cases in 2D and 3D as well as on microscopic images from mouse, ascidian, and *C. elegans* embryos. This computational framework significantly advances the previous attempts in the literature, which are appropriately referenced. The manuscript is well-written and this computational framework will be a great resource for the community. The manuscript would benefit by addressing the following comments:

1. Comment on why are the errors of contact angles so large (15-30 degrees) even for the very simple artificial examples in Fig. 2b. Also explain what is the meaning of the shaded region.
2. Comment on why are the relative errors of tension and pressure inference so large (more than 10%) even for the very simple artificial examples in Fig. 3e,f and Extended Data Fig. 3b,c. Also explain what statistical measures are represented with box-whisker plots.
3. The Supplementary Note does not provide enough details about how the variants of Young-Dupre were used to infer surface tensions. The captions of Extended Data Fig. 1a mention that the contact angles were calculated as the mean of dihedral angles of triplets of triangles along each junction, but this was never discussed in the Supplementary Notes. The authors should also comment on how they numerically minimized the weighted least square in Eq. (42). Ideally, authors should provide a similar level of detail as was done for the mesh-based variational force balance in section B.4.
4. The Supplementary Note does not provide enough details about how the Young-Laplace equations were used to infer pressures. Section A.4.5. only provides an expression for the local mean curvature at a given point. Are these mean curvatures of points then averaged over the whole interface between cells? Are Young-Laplace equations also weighted with the interface areas? Ideally, authors should provide a similar level of detail as was done for the mesh-based variational force balance in section B.4.

5. Comment on how uniform were the extracted values of surface tensions γ_{cm} and γ_{cc} for different cells in the 8-cell mouse embryos presented in Fig.3g
6. Inference of surface tensions from 64-cell embryos in Fig. 4 and Extended Data Fig. 4 suggested that mitotic blastomeres have higher apical tension than their interphase neighbors. Could this prediction be tested in experiments?
7. Text on page 10 states that the percentage of unstable junctions remains around 3% throughout the development of the ascidian embryo, but this is not evident from Fig. 4b.
8. I would like to alert the authors to another relevant preprint by Marin-Llaurado et al. titled “Mapping mechanical stress in curved epithelia of designed size and shape” on bioRxiv 2022.05.03.490382. This preprint should be cited as well.
9. Unfortunately, I was not able to test the code due to installation errors. Installation failed because of the robust-laplacian package. This prevented the dw3d and foambryo modules to build. I am using Mac OS Ventura and I tried to install the code in a clean conda environment.
10. Section B.2.3. discusses the Lagrangian function for embryos with line tensions, but it was not clear if this formulation was used anywhere in the manuscript. Line tensions were just briefly mentioned in the Discussion section of the main text.
11. Provide a reference for the Batchelor formula in the captions for Fig. 1i. Reference is only provided in the Supplementary Note.
12. Explain what is the meaning of matrices in Fig. 3d. Are these matrices providing values of surface tensions? Also provide colorbars.
13. In Fig. 4b it should be clearly marked that the axis displaying relative unstable junction length is plotted in percentages (%). Otherwise, the readers may think that the relative error is larger than 40%.
14. In Fig 4b, it is unclear what is plotted on the left axis. The figure indicates the Young-Dupre law residual, but captions and text refer to the percentage of unstable junctions in the embryo. If the latter is true, then again clearly indicate that the axis is showing percentage (%).
15. Captions of Fig. 4a say that mechanically unstable (stable) junctions are plotted in red (green) color. This should be moved to the description of the panel (c), where the relevant data is presented.
16. In Fig. 5c the colormap (red -> blue) is not consistent with the text and other colormaps (blue -> red)

17. Discussion on page 12 states that their pipeline achieves maximum relative force errors of around 11%. It is unclear where this number is coming from. Is this related to the table in Extended Data Fig. 3c? This should be made clear.

18. In section A.3.4. it was unclear how the weights were calculated in 3D. Were they averaging the value of the Euclidean distance transform like in 2D?

19. In Eq. (10) in Sec. A.4.4. the contact angle should be θ_b according to the Extended Data Fig. 1a.

20. Comment that Eq. (37) in section B.4.3. is used to set the average surface tension to 1.

21. Inconsistencies:

- matrices G_P vs G^P vs G^p
- areas A vs S , A_t vs S_t
- contact angles α vs θ

22. Typos:

- Page 5: text refers to Fig. 1a, but this should be Fig. 1d
- Page 8: text refers to Fig. 4b, but this should be Fig. 4d
- Page 10: missing norm $||\cdot||$ for residuals $A_\Gamma \times \Gamma - b_\gamma$
- Page 12: subsequent \rightarrow subsequent
- Page 19: there should be a negative sign in the second line of Eq. (7)
- Extended data Fig. 1d: $N_1(j)$ should be $N_1(i)$
- In the paragraph below Eq. (13) in the Supplemental Notes, the cell index should be 'k' and not 'l'
- Parentheses are missing in Eq. (15) in the Supplemental Notes
- Sections B.4.1 and B.4.2: $n_m \rightarrow n_m$, $n_c \rightarrow n_c$
- Missing norm $||\cdot||$ in the sentences above Eqs. (41) and (42) in the Supplemental Notes

Author Rebuttal to Initial comments

Response to Referees

Reviewers' Comments:

Reviewer #1:

Remarks to the Author:

The present manuscript presents an analysis pipeline and algorithm for force-inference measurements in 3D. It includes an advanced and superior mesh reconstruction method and a robust inversion algorithm. The method is tested on active foam simulations and on data of early embryos of mice, *C. elegans* and ascidian. Importantly, all in 3D, which is far from trivial. The article is clearly written, and the obtained precision of analysis is pretty impressive. The proper use of force inference technology is partly hampered by the lack of such beautiful paper and code. I am convinced that this manuscript will have a huge impact in the field and helps to perform more quantitative analysis in 3D cellular systems.

We thank the reviewer for her/his very positive appreciation of our work and we also hope that the method may be useful to the community.

Reviewer #2:

Remarks to the Author:

This manuscript represents a breakthrough. The work it describes is serious, systematic, validated and clearly explained. After suitable revision and polishing, it will be instrumental in making 3D stress inference a mature technique of great interest to investigate living tissue stresses in situ, in a non-invasive way, and with a high throughput.

We thank the reviewer for her/his positive appreciation of our work.

* Vocabulary :

- Keep "surface tension" for the interface between two mediums or tissues (eg the whole embryo "approximated to a droplet"). Here use "cell-cell tension", "cell-cell junction tension", "junction tension" or any other precise term.

We think that 'surface tension' remains the best term to generically refer to the tensions we infer in this work: cell-cell or junction tensions would not be appropriate to refer to cell-medium tensions (sometimes called cortical tensions) for instance. We better defined the terms at the beginning of the manuscript and kept the same ones throughout the paper.

- "Pressure inference" and "Tension inference" are correct terms. But "Force inference", or "inference" alone, should be replaced with "stress inference" (see Noll 2020), "cell stress inference", "tension and pressure inference" or any other precise term.

Noll et al. PRX 2020 use alternatively 'force inference' and 'stress inference', as well as 'forces' and 'stresses' in their manuscript. Here pressures are homogeneous to a stress, surface tensions are homogeneous to a force per unit length, so we preferred 'mechanical inference' as generic term to describe the process of solving an inverse mechanical problem (e.g. retrieving a force, stress or tension) and we used 'tension and pressure inference' to refer specifically to our method.

- The term "active foam", applied to a living tissue, underlines the analogy and differences between a foam and a tissue. It has been popularized in a context where it was correct: a discussion of the effect of cell ATP consumption on the analogy validity. Here the difference between the tissue and the foam lies in the fact that tension varies from one cell junction to another. The effects of this spatial heterogeneity are completely independent of its origin (whether active or not) and the current manuscript can be applied to passive heterogeneous foams too. Hence replace "active foam", which is ambiguous, with "heterogeneous foam" or any other precise term.

We replaced everywhere 'active foam' model by 'foam-like' model. We made explicit in the beginning that it corresponds to a heterogeneous foam.

- The Surface Evolver is not a "vertex model".

- In addition, The Surface Evolver can be called a simulation "on a discrete mesh" but this can create confusion with simulations on a discrete lattice. Preferably call it a simulation "of discretised junctions", "of triangulated junctions" or any other precise term.

Surface Evolver might not be thought by the community as a 'vertex model' if one refers to the models popularized by the manuscript of Farhadifar et al. 2007 or the 3D variants proposed earlier (works by H. Honda or even before by G. Odell) or more recently (works by E. Hannezo or G. Salbreux). Yet, it is fundamentally a model based on a discrete approximation of the geometry with vertices and the optimization of the positions of such vertices from an energy, which is the basis of all vertex models previously cited. Actually, an equivalent model to Surface Evolver in 2D was called genuinely "bubbly vertex model" (Ishimoto et al. 2014). In contrast, Surface Evolver is not a finite-element model, which would require quadrature rules on the mesh that parametrizes the surface. The only difference between Surface Evolver (or equivalently our simulations, which are similar but automatically handle topology transitions) with what the author - and generally the community - refers to as 'vertex models' lies in the fact that interfaces are not arbitrarily supposed flat, requiring more degrees of freedom. In fact, we believe that Surface Evolver (or our equivalent simulations) falls exactly within the general class of vertex models, in contrast to other approaches. We modified nevertheless the text as we realize that our point may deserve a thorough discussion, that does not fit within this manuscript.

* Other :

In introduction, discuss Noll's approach to simultaneously segment the pattern and equilibrate it in 2D ; then Liu, Lemaire et al's preprint attempting at doing the same in 3D, and its limitations ; and then finally how the current manuscript is positioned. The exact approach should be better explained, underlining what is standard, what is new, the number of cells which can be treated, and what validations are performed.

We followed the Reviewer's proposition regarding the presentation order in the Introduction and we thank her/him for her/his advice. We want to keep the discussion on Liu et al.'s preprint minimal, as we don't believe our manuscript is the right place to comment in details the results of a manuscript that builds on a mathematical assumption that is fundamentally incorrect.

Reviewer #3:

Remarks to the Author:

The authors presented a computational framework that uses microscopic images of tissues to generate 3D meshes, which are then used to estimate surface tensions and cell pressures. The computational framework was tested on a few simple artificial cases in 2D and 3D as well as on microscopic images from mouse, ascidian, and C. elegans embryos. This computational framework significantly advances the previous attempts in the literature, which are appropriately referenced. The manuscript is well-written and this computational framework will be a great resource for the community. The manuscript would benefit by addressing the following comments:

We thank the Reviewer for her/his positive appreciation of our work and the very detailed and careful report on our manuscript. However, we have not tested our framework in 2D, and focus on the 3D inference.

1. Comment on why are the errors of contact angles so large (15-30 degrees) even for the very simple artificial examples in Fig. 2b.

We thank the Reviewer for pointing out this important aspect: this error is intrinsic to our segmentation method. The reconstruction of geometry being based on local image information (no smoothing or regularization was added so that we do not bias the geometry) and is therefore directly limited by the image resolution. To better quantify this aspect, we started from a mesh example and systematically varied the resolution of the artificially generated image. We calculated the relative error in geometry reconstruction (angles), and the consecutive relative error in inferred tensions with Young-Dupré and pressures with our variational Laplace formula. The plots are shown below, illustrating - as one may expect - that a main factor limiting geometry reconstruction is the original image resolution. We commented this important aspect in the manuscript (page 8) and added the plots as Extended Fig. 3c.

While we demonstrated that our Delaunay-watershed mesh generation approach is performing better than state-of-the-art algorithms for geometry reconstruction (CGAL), we acknowledge that there is still room for improvement in that direction. We believe that active mesh segmentation methods based on differential microscopy image rendering - although computationally much more costly - may offer an alternative but efficient solution (Ichbiah, Delbary & Turlier [arXiv:2303.10440](https://arxiv.org/abs/2303.10440) 2023).

Also explain what is the meaning of the shaded region.

The shaded region corresponds to the standard deviation: we added this point in the corresponding legend. We display these details in the Methods section, in the paragraph Statistical Analysis.

2. Comment on why are the relative errors of tension and pressure inference so large (more than 10%) even for the very simple artificial examples in Fig. 3e,f and Extended Data Fig. 3b,c. Also explain what statistical measures are represented with box-whisker plots.

We thank the Reviewer for her/his question. We actually don't think that a relative error of ~10% on inferred tensions/pressures starting from raw images is very large: it is in the same range - if not better - than the typical error previously obtained with 2D inference methods, where geometry reconstruction is a simpler task (Kong et al. Scientific Report 2019, Noll et al PRX 2020). As we illustrated above, this error on inferred tension/pressure values comes intrinsically from imprecision in the reconstructed geometry. If we started from simulated meshes directly (and not reconstructed ones from artificial images), this error would be negligible. In the manuscript, we furthermore used images of similar sizes (about 250² pixels) for the benchmarking of our

method. As we showed above, the inference error is expected to decrease with images of larger sizes (at the same typical number of cells). We now refer explicitly to this aspect in the manuscript (page 8).

The box center is located at the median, and its extremities represents the first and third quartiles. The whiskers are located at $Q1 - 1.5*(Q3-Q1)$ and $Q3 + 1.5*(Q3-Q1)$. We added these details in the Methods section, in the paragraph Statistical Analysis.

3. The Supplementary Note does not provide enough details about how the variants of Young-Dupre were used to infer surface tensions. The captions of Extended Data Fig. 1a mention that the contact angles were calculated as the mean of dihedral angles of triplets of triangles along each junction, but this was never discussed in the Supplementary Notes. The authors should also comment on how they numerically minimized the weighted least square in Eq. (42). Ideally, authors should provide a similar level of detail as was done for the mesh-based variational force balance in section B.4.

We thank the Reviewer and added more details in the Supplementary note to explain these points (see Supplementary Note section A.4.4, page 4, section B.3, page 6, and section C.3, page 10).

4. The Supplementary Note does not provide enough details about how the Young-Laplace equations were used to infer pressures. Section A.4.5. only provides an expression for the local mean curvature at a given point. Are these mean curvatures of points then averaged over the whole interface between cells? Are Young-Laplace equations also weighted with the interface areas? Ideally, authors should provide a similar level of detail as was done for the mesh-based variational force balance in section B.4.

We thank the Reviewer and added more details in the Supplementary Note to explain these points. (see Supplementary Note section A.4.5, page 4, and section C.5.3, page 11).

5. Comment on how uniform were the extracted values of surface tensions γ_{cm} and γ_{cc} for different cells in the 8-cell mouse embryos presented in Fig.3g

We thank the Reviewer for the suggestion. We calculated the standard deviation of γ_{cm} and γ_{cc} among cells of same embryos and we compared it to published values (Maitre et al. Nat Cell Biol 2015) in the Extended Figure 3d.

6. Inference of surface tensions from 64-cell embryos in Fig. 4 and Extended Data Fig. 4 suggested that mitotic blastomeres have higher apical tension than their interphase neighbors. Could this prediction be tested in experiments?

We thank the Reviewer for the suggestion. Measuring apical surface tension on blastomeres in the ascidian embryo becomes extremely tedious from the 64-cell stage onward because a pipette of much smaller size has to be used, which generally leads to blebbing and therefore wrong tension measurements. Such experiment has been tried by the authors but also other colleagues in the past, no one managed to solve this issue so far, explaining why no tension measurement is available in the literature from this stage in ascidian embryos. Circumventing this problem is out-of-scope for this manuscript, but we are working on a way to circumvent this issue to confirm experimentally the new prediction made by our method.

7. Text on page 10 states that the percentage of unstable junctions remains around 3% throughout the development of the ascidian embryo, but this is not evident from Fig. 4b.

We thank the Reviewer for pointing out that the corresponding data is missing. We added the corresponding figure in the blue plot of Fig. 4b, and plotted the value of the Young-Dupré equations residuals across development in Fig. 4c.

8. I would like to alert the authors to another relevant preprint by Marin-Llaurado et al. titled "Mapping mechanical stress in curved epithelia of designed size and shape" on bioRxiv 2022.05.03.490382. This preprint should be cited as well.

We thank the Reviewer for pointing out this paper and we added it to the references (page 1).

9. Unfortunately, I was not able to test the code due to installation errors. Installation failed because of the robust-laplacian package. This prevented the dw3d and foambryo modules to build. I am using Mac OS Ventura and I tried to install the code in a clean conda environment.

We are sorry to hear this and we updated the code to remove the robust-Laplacian module, which may indeed cause some installation issue. The code is now installable directly via pip and we tested on several machines with clean MacOS Ventura installations (with native Python) and Linux Manjaro: <https://pypi.org/project/delaunay-watershed-3d>
<https://pypi.org/project/foambryo>

10. Section B.2.3. discusses the Lagrangian function for embryos with line tensions, but it was not clear if this formulation was used anywhere in the manuscript. Line tensions were just briefly mentioned in the Discussion section of the main text.

We are sorry for this mistake. Our formulation does not allow for line tension inference at this point and we have removed this section.

11. Provide a reference for the Batchelor formula in the captions for Fig. 1i. Reference is only provided in the Supplementary Note. We thank the Reviewer for pointing this out and we added the reference in the main text as well.

12. Explain what is the meaning of matrices in Fig. 3d. Are these matrices providing values of surface tensions? Also provide colorbars.

We thank the Reviewer for pointing this out. We added a colorbar for the tension values and extended the corresponding legend.

13. In Fig. 4b it should be clearly marked that the axis displaying relative unstable junction length is plotted in percentages. Otherwise, the readers may think that the relative error is larger than 40%.

We thank the Reviewer for pointing this out and we corrected it.

14. In Fig 4b, it is unclear what is plotted on the left axis. The figure indicates the Young-Dupre law residual, but captions and text refer to the percentage of unstable junctions in the embryo. If the latter is true, then again clearly indicate that the axis is showing percentage.

We thank the Reviewer for pointing this out. This was indeed the residual of the inference. We plot now the value of the Young-Dupré equations residual across development in Fig. 4c and the percentage of unstable junctions in the embryo in Fig.4b.

15. Captions of Fig. 4a say that mechanically unstable (stable) junctions are plotted in red (green) color. This should be moved to the description of the panel (c), where the relevant data is presented.

We corrected it.

16. In Fig. 5c the colormap (red -> blue) is not consistent with the text and other colormaps (blue -> red)

We corrected it.

17. Discussion on page 12 states that their pipeline achieves maximum relative force errors of around 11%. It is unclear where this number is coming from. Is this related to the table in Extended Data Fig. 3c? This should be made clear.

We thank the Reviewer, this % comes indeed from this table. This table is now moved to the Supplementary Note section C.4.2, page 11, and we now refer explicitly to it in the text (page 11).

18. In section A.3.4. it was unclear how the weights were calculated in 3D. Were they averaging the value of the Euclidean distance transform like in 2D?

We thank the Reviewer and corrected it in the Supplementary.

19. In Eq. (10) in Sec. A.4.4. the contact angle should be θ_b according to the Extended Data Fig. 1a.

We thank the Reviewer and corrected it.

20. Comment that Eq. (37) in section B.4.3. is used to set the average surface tension to 1.

We thank the Reviewer and corrected it.

21. Inconsistencies:

- matrices G_P vs G^P vs G^p
- areas A vs S , A_t vs S_t
- contact angles α vs θ

We thank the Reviewer and corrected these inconsistencies.

22. Typos:

- Page 5: text refers to Fig. 1a, but this should be Fig. 1d
- Page 8: text refers to Fig. 4b, but this should be Fig. 4d
- Page 10: missing norm $|| \cdot ||$ for residuals $A_\Gamma \times \Gamma - b_\gamma$
- Page 12: subsequent -> subsequent
- Page 19: there should be a negative sign in the second line of Eq. (7)
- Extended data Fig. 1d: $N_1(j)$ should be $N_1(i)$
- In the paragraph below Eq. (13) in the Supplemental Notes, the cell index should be 'k' and not 'l'
- Parentheses are missing in Eq. (15) in the Supplemental Notes
- Sections B.4.1 and B.4.2: $n_m \rightarrow n_m$, $n_c \rightarrow n_c$
- Missing norm $|| \cdot ||$ in the sentences above Eqs. (41) and (42) in the Supplemental Notes

We thank the Reviewer and corrected these typos.

Decision Letter, first revision:

Dear Herve,

Thank you for submitting your revised manuscript "Embryo mechanics cartography: inference of 3D force atlases from fluorescence microscopy" (N METH-A52264A). It has now been seen by the original referees and their comments are below. The reviewers find that the paper has improved in revision, and therefore we'll be happy in principle to publish it in Nature Methods, pending minor revisions to satisfy the referees' final requests and to comply with our editorial and formatting guidelines.

TRANSPARENT PEER REVIEW

Nature Methods offers a transparent peer review option for new original research manuscripts submitted from 17th February 2021. We encourage increased transparency in peer review by publishing the reviewer comments, author rebuttal letters and editorial decision letters if the authors agree. Such peer review material is made available as a supplementary peer review file. Please state in the cover letter 'I wish to participate in transparent peer review' if you want to opt in, or 'I do not wish to participate in transparent peer review' if you don't. Failure to state your preference will result in delays in accepting your manuscript for publication.

ORCID

Sincerely,
Madhura

Madhura Mukhopadhyay, PhD
Senior Editor
Nature Methods

Reviewer #3 (Remarks to the Author):

Authors have addressed all of my previous concerns and I recommend publication. I was also able to install and try the software package this time, which will be a great resource for the community.

Minor comments:

- 1.) Authors still haven't explained how they minimized weighted least-squares functions in sections C.4.2 and C.5.3. Authors only commented on how they solved the ordinary least-squares in section C.3.
- 2.) Page 6: inconsistency for the number of cells n_c vs n_C
- 3.) Page 8: sentence discussing the systematically lower variability in inferred γ_{cc} values should refer to the Extended data Fig. 3d instead of Fig. 3d
- 4.) Page 10: typo (Fig 4. 4e)
- 5.) Page 12: sentence discussing the relative force errors of about 10% should refer to the tables in sections C.4.2 and C.5.3 and not to Fig. 3c.
- 6.) Page 23: Captions are missing the description for the table in panel (d) in the Extended data Fig. 3
- 7.) Sections A.4.1, A.4.2, A.4.5: there are still some inconsistencies in the labels of cell areas (A vs S)
- 8.) Section B.4.4: inconsistency G_P vs G^p

Author Rebuttal, first revision:

Response to Referees

Reviewers' Comments:

Reviewer #3:

Remarks to the Author:

Authors have addressed all of my previous concerns and I recommend publication. I was also able to install and try the software package this time, which will be a great resource for the community.

Response: We are thankful to the Referee for her/his positive appreciation of our work and for the very careful reviewing work.

Minor comments:

1.) Authors still haven't explained how they minimized weighted least-squares functions in sections C.4.2 and C.5.3. Authors only commented on how they solved the ordinary least-squares in section C.3.

Response: we solve the weighted least-square functions in sections C.4.2 and C.5.3 exactly the same way. We added this information in the Supplementary Note.

2.) Page 6: inconsistency for the number of cells n_c vs n_C

Response: this is corrected.

3.) Page 8: sentence discussing the systematically lower variability in inferred γ_{cc} values should refer to the Extended data Fig. 3d instead of Fig. 3c

Response: this is corrected.

4.) Page 10: typo (Fig 4. 4e)

Response: this is corrected.

5.) Page 12: sentence discussing the relative force errors of about 10% should refer to the tables in sections C.4.2 and C.5.3 and not to Fig. 3c.

Response: this is corrected.

6.) Page 23: Captions are missing the description for the table in panel (d) in the Extended data Fig. 3

Response: this is corrected.

7.) Sections A.4.1, A.4.2, A.4.5: there are still some inconsistencies in the labels of cell areas (A vs S)

Response: this is corrected.

8.) Section B.4.4: inconsistency G_P vs G^p

Response: this is corrected.

Final Decision Letter:

Dear Hervé,

I am pleased to inform you that your Article, "Embryo mechanics cartography: inference of 3D force atlases from fluorescence microscopy", has now been accepted for publication in Nature Methods. Your paper is tentatively scheduled for publication in our December print issue, and will be published online prior to that. The received and accepted dates will be 13 Apr, 2023 and 12 Oct, 2023. This note is intended to let you know what to expect from us over the next month or so, and to let you know where to address any further questions.

Over the next few weeks, your paper will be copyedited to ensure that it conforms to Nature Methods style. Once your paper is typeset, you will receive an email with a link to choose the appropriate publishing options for your paper and our Author Services team will be in touch regarding any additional information that may be required.

You will receive a link to your electronic proof via email with a request to make any corrections within 48 hours. If, when you receive your proof, you cannot meet this deadline, please inform us at rjsproduction@springernature.com immediately.

Please note that *Nature Methods* is a Transformative Journal (TJ). Authors may publish their research with us through the traditional subscription access route or make their paper immediately open access through payment of an article-processing charge (APC). Authors will not be required to make a final decision about access to their article until it has been accepted. [Find out more about Transformative Journals](https://www.springernature.com/gp/open-research/transformative-journals)

Authors may need to take specific actions to achieve [compliance](https://www.springernature.com/gp/open-research/funding/policy-compliance-faqs) with funder and institutional open access mandates. If your research is supported by a funder that requires immediate open access (e.g. according to [Plan S principles](https://www.springernature.com/gp/open-research/plan-s-compliance))

then you should select the gold OA route, and we will direct you to the compliant route where possible. For authors selecting the subscription publication route, the journal's standard licensing terms will need to be accepted, including [self-archiving policies](https://www.springernature.com/gp/open-research/policies/journal-policies). Those licensing terms will supersede any other terms that the author or any third party may assert apply to any version of the manuscript.

Your paper will now be copyedited to ensure that it conforms to Nature Methods style. Once proofs are generated, they will be sent to you electronically and you will be asked to send a corrected version within 24 hours. It is extremely important that you let us know now whether you will be difficult to contact over the next month. If this is the case, we ask that you send us the contact information (email, phone and fax) of someone who will be able to check the proofs and deal with any last-minute problems.

If, when you receive your proof, you cannot meet the deadline, please inform us at rjsproduction@springernature.com immediately.

Once your manuscript is typeset and you have completed the appropriate grant of rights, you will receive a link to your electronic proof via email with a request to make any corrections within 48 hours. If, when you receive your proof, you cannot meet this deadline, please inform us at rjsproduction@springernature.com immediately.

Once your paper has been scheduled for online publication, the Nature press office will be in touch to confirm the details.

Once your paper has been scheduled for online publication, the Nature press office will be in touch to confirm the details.

Content is published online weekly on Mondays and Thursdays, and the embargo is set at 16:00 London time (GMT)/11:00 am US Eastern time (EST) on the day of publication. If you need to know the exact publication date or when the news embargo will be lifted, please contact our press office after you have submitted your proof corrections. Now is the time to inform your Public Relations or Press Office about your paper, as they might be interested in promoting its publication. This will allow them time to

prepare an accurate and satisfactory press release. Include your manuscript tracking number NMETH-A52264B and the name of the journal, which they will need when they contact our office.

About one week before your paper is published online, we shall be distributing a press release to news organizations worldwide, which may include details of your work. We are happy for your institution or funding agency to prepare its own press release, but it must mention the embargo date and Nature Methods. Our Press Office will contact you closer to the time of publication, but if you or your Press Office have any inquiries in the meantime, please contact press@nature.com.

Nature Portfolio journals [encourage authors to share their step-by-step experimental protocols](https://www.nature.com/nature-research/editorial-policies/reporting-standards#protocols) on a protocol sharing platform of their choice. Nature Portfolio 's Protocol Exchange is a free-to-use and open resource for protocols; protocols deposited in Protocol Exchange are citable and can be linked from the published article. More details can found at www.nature.com/protocolexchange/about.

Best regards,
Madhura

Madhura Mukhopadhyay, PhD

Senior Editor
Nature Methods